# Batchnorm Allows Unsupervised Radial Attacks

**Amur Ghose**[*1,2], **Apurv Gupta**[3], **Yaoliang Yu**[1,2], **Pascal Poupart**[1,2]
[1]David R. Cheriton School of Computer Science, University of Waterloo,
[2]Vector Institute, [3]Columbia University
a3ghose@uwaterloo.ca, apurvgupta1996@gmail.com,
yaoliang.yu@uwaterloo.ca,ppoupart@uwaterloo.ca

## Abstract

The construction of adversarial examples usually requires the existence of soft or hard labels for each instance, with respect to which a loss gradient provides the signal for construction of the example. We show that for batch normalized deep image recognition architectures, intermediate latents that are produced after a batch normalization step by themselves suffice to produce adversarial examples using an intermediate loss solely utilizing angular deviations, **without** relying on any label. We motivate our loss through the geometry of batch normed representations and their concentration of norm on a hypersphere and distributional proximity to Gaussians. Our losses expand intermediate latent based attacks that usually require labels. The success of our method implies that leakage of intermediate representations may create a security breach for deployed models, which persists even when the model is transferred to downstream usage. Removal of batch norm weakens our attack, indicating it contributes to this vulnerability. Our attacks also succeed against LayerNorm empirically, thus being relevant for transformer architectures, most notably vision transformers which we analyze.

## 1 Introduction

Adversarial examples $x_{adv}$ are commonly defined as data instances which lie $\epsilon$ away in some norm (usually $L_\infty$) from an actual data instance $x_{real}$. To humans, $\epsilon$ is small and $x_{real}, x_{adv}$ share labels, yet to a classifier neural network, they do not. Since their discovery [Szegedy et al., 2013, Goodfellow et al., 2014], adversarial examples have spurred research in both attacking (i.e., generating adversarial data instances) [Akhtar and Mian, 2018, Tramer et al., 2020] and defending [Yuan et al., 2019] neural networks against them. Common methods for generating adversarial images use loss gradients. A pioneering attack is Fast Gradient Sign Method (FGSM) [Goodfellow et al., 2014], possibly strengthened via Projected Gradient Descent (PGD) [Madry et al., 2017] - an iterative form of FGSM. Defenses against these attacks have been derived, including adversarial training [Shrivastava et al., 2017], where adversarial images are fed to the network for training. Adversarial training is expensive but powerful [Salman et al., 2019] and scalable [Wong et al., 2020, Shafahi et al., 2019]. Defenses can give illusory security by obfuscating gradients [Athalye et al., 2018]. An unsaid aspect of defense is hiding model parameters - changing attacks from the **white-box** (model parameters known) scenario to black-box (models only accessible via outputs) scenario, which makes them harder to attack.

We consider the family of **intermediate-level** attacks, exemplified by members such as Intermediate Level Attack Projection (ILAP) [Huang et al., 2019] and its variants [Li et al., 2020b]. In this, an initial direction in the latent space of the hidden intermediate layers is determined via FGSM or other attacks. Gradient steps then maximally perturb the intermediate hidden representation to find a suitable $x_{adv}$, unlike directly using the label. Such methods can outperform the method used for

---

[*]Corresponding author: Amur Ghose, email: a3ghose@uwaterloo.ca

the initial direction itself and be more transferable across different models. ILAP uses FGSM to set initial directions, but once the direction arises, the layers after the intermediate layer play no role.

We now ask : Given an unlabeled data point $x$ along with full access to a deep network upto $K$ layers, where $K$ is less than the network depth, could we craft an adversarial example out of $x$ without any other information - e.g., the label of $x$, any $x' \neq x$ - so long as the deep network utilized batch normalization ? We answer this question positively when $x$ is an image for several popular families of image recognition architectures, exploiting the hyperspherical geometry of latent spaces to propose an **angular adversarial attack.** The label access of the black box case is absent - we do not allow access to hard labels (true label of $x$) or soft labels (logits obtained at penultimate layers). We adapt the method advanced in ILAP to remove the need for label-based FGSM, and proceed instead with an angular loss based on the geometry of batch normalization. **Our concrete contributions are** :

- To provide a label-free attack that relies on only gradient access upto intermediate layers
- Ablation of our method to show it is not brittle to choices of intermediate layers, (small variations between using layers $i, i + 1$ if $i$ is sufficiently deep)
- To show prominent vision models - ResNets and EfficientNets - fall to our attack. Removing batchnorm in ResNets via Fixup neuters the attack - suggesting it is playing a role in this.
- To show that the attack **empirically works even for transformer architectures** (ViT) that employ Layernorm over batchnorm.
- To show that these attacks persist in the transfer learning setting, where the models are frozen upto a certain layer and then fine-tuned for downstream usage.
- To improve the supervised attack case when labels are available by using our loss in conjunction with the loss from the true label.

This implies that releasing the first $K$ layers of a model publicly as a feature extractor, without any label access, can create adversarial attacks, which has security implications. Weaknesses persist even when the unseen, later layers of the network are tuned on different datasets (e.g. CIFAR10/100) - adversarial examples can be crafted for CIFAR without using the changed, later layers only using this first $K$. We exploit the radial geometry of batch normalization and assume the norm of the neural representations concentrates around a constant, allowing us to exploit the geometry formed. This indicates batch normalization may make networks not just less robust to conventional attacks [Galloway et al., 2019], but open up new attacks entirely. We ablate our method and compare to scenarios of naive baselines and having access to $x' \neq x$ to showcase its competitiveness.

## 1.1   Relation to Related Work

We tie together two areas : analyzing batch normalization from a geometric standpoint under convergence assumptions, and adversarial attacks with partial knowledge of architectures. Batch normalization has been analyzed from the standpoint of deep learning theory and model optimization [Santurkar et al., 2018]. While its detrimental effects on model robustness [Galloway et al., 2019, Benz et al., 2020] are known, most analysis focuses on how batch normalization makes the model weaker to already-existing attacks. We propose that batch normalization might impact the network by making it more vulnerable to novel attacks. We are to our knowledge the first to connect this with the concentration of norm phenomenon implied by results such as the dynamics of batch normalization at high depths [Daneshmand et al., 2021]. These tie into results such as neural collapse [Papyan et al., 2020], which find that later layers in deep networks have predictable geometric patterns.

Research in adversarial examples has focused in specific settings such as white/grey/black box settings denoting levels of model access/knowledge. Even in the black box case, it is often assumed we can feed an input $x$ into the model $M$ under attack and retrieve $M(x)$, the model's prediction of the label of $x$. We consider the case of partial model knowledge and attacking a model with no label outputs. This is relevant with the presence of large-scale pretraining, where models are fed large training sets of data unrelated to any one domain to capture the overall distribution of a particular type of modality, such as GPT-3 [Brown et al., 2020] with natural language. Deployed downstream, these public models may fine tune on a smaller dataset, and most of their weights are not fine-tuned. Public models may act as feature extractors that distill large amounts of data beyond dataset scopes. They output not labels, but representation vectors, and our attack targets this case. Previous works do use latent representations to construct adversarial examples [Park and Lee, 2021, Creswell et al.,

2017, Yu et al., 2021] through moving in latent space towards points of differing labels or maximizing deviations in latent space [Kos et al., 2018]. Along with the related **no-box** [Li et al., 2020a, Zhang et al., 2022b] setting (assuming no model knowledge), these methods require extra parts, such as representations of instances of different labels as targets to move towards, surrogate models, or some soft labels, often with gradient access - all of which we do not require. They are also ad hoc in their choice of layers to attack, but we give a definite answer of which layers and architectures are the most vulnerable - batch norm layers situated deeper in the network. The closest case to our attack are label-free, single-stage latent-based attacks [Zhou et al., 2018, Inkawhich et al., 2019, Ganeshan et al., 2019, Inkawhich et al., 2020b, Zhang et al., 2022a, Wang et al., 2021, Inkawhich et al., 2020a] which we outperform. These methods use the latent to boost the supervised scenario and not exploit batch normalization or two-step optimization in an unsupervised setting as we do.

## 1.2 Setup - FGSM and PGD Attacks

We describe FGSM aka Fast Gradient Sign Method [Goodfellow et al., 2014] for generating an adversarial example. The task is to find $x'$ which differs from $x$ in the label assigned by a classifier $C$, which outputs a distribution over $K$ classes ($K$ positive numbers that sum to 1).

$$\|x' - x\| \leq \epsilon \qquad \text{and}$$
$$\arg\max_i [C(x)]_i \neq \arg\max_i [C(x')]_i \tag{1}$$

Generally, we only restrict attacks to cases where $\arg\max_i [C(x)]_i$ is the true label, i.e., $C$ is correct. The norm $\|.\|$ used to define the attack is almost always the $L_\infty$ norm, though other choices such as $L_2$ [Carlini and Wagner, 2017] have been tested as well. We will assume the norm to be $L_\infty$ for the description. Computing $x'$ (possibly with projection to a valid domain) proceeds as follows:

- Apply $C$ to $x$ to obtain $C(x)$.
- Compute gradient $\nabla L$ wrt $x$ where $L$ is the loss between $C(x)$ and the true label vector $v_l$.
- Compute $x'_i = x_i + \epsilon \times \text{sign}(\nabla L)_i$, i.e., move by $\pm \epsilon$ based on sign of gradient.

We understand the **sign** function to map to $-1, 0, 1$ respectively as the argument is $< 0, 0,$ or $> 0$. FGSM uses gradient information, which may vary rapidly around $x$. A stronger variant, the Projected Gradient Descent (PGD) attack [Madry et al., 2017] uses multiple steps near $x$ taking gradients at each step. Under PGD, there is a step size $\alpha$, typically $<< \epsilon$, which iteratively updates $x_i^t$ as :

$$x_i^t = x_i^{t-1} + \alpha \times \text{sign}(\nabla L)_i^{t-1}$$

With a clamping step on $x_i^t$ that enforces the constraint of $\|x' - x\| \leq \epsilon$, this step after initializing $x_i^0$ at $x_i$ yields much stronger adversarial examples than FGSM at higher $t$.

## 1.3 Intermediate Level Attack Projection (ILAP)

These methods - FGSM and PGD - rely on the input and the final layer's output loss with respect to the true label. We can consider a neural network $N$ of depth $D$ as :

$$N(x) = N_{i+1,D}(N_{1,i}(x))$$

With 1 based indexing, $N_{j,k}$ is the neural sub-network via composing layers from depth $j$ to $k$. Let $z_i = N_{1,i}(x)$ be the latent representation at depth $i$. Consider $x_{adv}$ from any baseline method, and let

$$z_i^{adv} = N_{1,i}(x_{adv}) \; ; \; z_i^{orig} = N_{1,i}(x) \; ; \; \Delta_i^{adv} = z_i^{adv} - z_i^{orig}$$

We then seek an alternate adversarial example, $x_i^{ILAP}$ which seeks to work with the loss function :

$$L = \langle \Delta_i^{adv}, \Delta_i^{ILAP} \rangle \text{ where } \Delta_i^{ILAP} = z_i^{ILAP} - z_i^{orig} \text{ and } z_i^{ILAP} = N_{1,i}(x_i^{ILAP})$$

The $L$ can then be plugged into either FGSM or PGD. Put simply, this loss encourages latent space movement directionally similar to FGSM/PGD. $L$ increases the inner product with $\Delta_i^{adv}$ and later iterations use it over the true loss. This alternate adversarial example is highly transferable - e.g., adversarial examples from ResNet18 have higher success rates vs different models such as DenseNet, when created by an ILAP process than directly using FGSM. It can even beat FGSM on the source model itself. [Huang et al., 2019] ILAP demonstrates that latent representations provide ammunition for adversarial attacks, but requires an initial adversarial example $x_{adv}$ created with full model access and label. To remedy this, we craft an attack needing neither. We show that if the correct layers are attacked with an angular version of the same loss, the initial adversarial example is unnecessary.

## 2 Geometry of Batch Normalization

Given a minibatch of inputs $x_i$ of dimensions $n$ with mean $\mu_{ij}$ and standard deviation $\sigma_{ij}$ at index $j$, consider batch normalization [Ioffe and Szegedy, 2015] or just **BN** as:

$$[\text{BN}(x_i)]_j = \frac{x_{ij} - \mu_{ij}}{\sigma_{ij}} \tag{2}$$

Such normalization (with affine shift) has become commonplace in neural networks. During training, the empirical average of the minibatch is used and during testing, computed averages (mean/standard deviation) are used. By batchnorm, we mean only the batchnorm layer without an affine shift. Layer normalization [Ba et al., 2016] is an alternate form of normalization that normalizes over all neurons in a hidden layer. For space reasons, we discuss batch normalization in the main text and layer normalization in Appendix A. Empirically, our attack succeeds on layer norm and architectures using it, such as transformers [Vaswani et al., 2017] and vision transformers [Dosovitskiy et al., 2020].

**Converged regime:** Suppose in training, the sample means/standard deviations in a pre-BN layer converge to sample statistics. Post-batchnorm, representation vector $Z$ satisfies, at every index $j$:

$$\mathbb{E}(Z_j) = 0, \quad \mathbb{E}\left(Z_j^2\right) = 1 \tag{3}$$

i.e., for a layer of dimensionality $d$ and denoting the entire latent vector as $Z$, $E(Z) = 0, E(\|Z\|^2) = d$ by linearity of expectation. The above convergence is with respect to the **training** set, but carries over to the **test** set (approximately) under the I.I.D assumption when learning takes place. Suppose $\|Z\|^2$ was concentrated about its expected value - i.e., nearly all $\|Z\|$ values were $\approx \sqrt{d}$. We first discuss the implications of such a scenario and what attacks they allow. We assume that in this latent space, the inner product determines 'similarity'. Two latents $Z_a, Z_b$ from different instances $X_a, X_b$ will be closer as $\langle Z_a, Z_b \rangle$ rise. Under this metric , the most dissimilar point to $Z_a$ is parallel to $-Z_a$. We will formulate our attack assuming that $\|Z\| = \sqrt{d}$ i.e. a hyperspherical latent space. After we have formulated our attack, we consider scenarios of such spaces and concentrations of norm.

### 2.1 Angular Attack Based on Converged Batchnorm

Let $N_{1,i}$ be the network upto layer $i$ with dimensionality $d$. If layer $i$ is a batch norm layer (without affine shift), $z = N_{1,i}(x)$ lies (with converged batchnorm and $\|Z\|$'s norm concentrated) approximately on a shell of radius $\sqrt{d}$. The natural distance metric between two $z, z'$ is angular distance:

$$\frac{\langle z, z' \rangle}{\|z\|\|z'\|}$$

where $\langle z, z' \rangle$ is at a local maximum on the hypersphere $\|z\|$ = constant when $z = z'$ and no gradient exists. We need another gradient initially, and propose the following attack algorithm. Given a real example $x^0$ with latent $z^0 = N_{1,i}(x^0)$, $z^t = N_{1,i}(x^t)$ we consider the loss $L_{\text{init}}^t = -\|z^t\| = -\|N_{1,i}(x^t)\|$ for the first $t_{\text{init}}$ iterations. We then iteratively generate $x^t$:

$$x^t = x^{t-1} + \alpha \times \text{sign}\left(\nabla L_{\text{init}}^{t-1}\right) \quad ; \quad 1 \le t \le t_{\text{init}}$$

After $t_{\text{init}}$ iterations, we modify the angular loss to $L_{\text{radial}}$, using the deviations obtained so far. $L_{\text{radial}}$ is angularly defined, denoting $z_i^t$ the latent at depth $i$ and iteration $t$ :

$$L_{\text{radial}} = -\frac{\langle z_i^0, z_i^t \rangle}{\|z_i^0\|\|z_i^t\|}; z_i^0 = N_{1,i}(x^0), z_i^t = N_{1,i}(x^t)$$

We then iteratively update as follows (graphically depicted in Figure 1):

$$x^t = x^{t-1} + \alpha \times \text{sign}\left(\nabla L_{\text{radial}}^{t-1}\right) \quad ; \quad t_{\text{init}} < t \le t_{\text{radial}}$$

The figure geometrically depicts how intermediately moving from a point to another in latent space lowers the radial norm ($A_i$, left). Directly lowering the radial norm via a path from $A_0$ to $A_1$ (right) can create an initial deviation. But extending this to $C$ as in the figure ends up with a lower angular deviation than maximizing angular deviation directly and ending up at $A_{adv}$. Methods which directly increase the $L_2$ deviation starting from an initial method would result in points such as $C$.

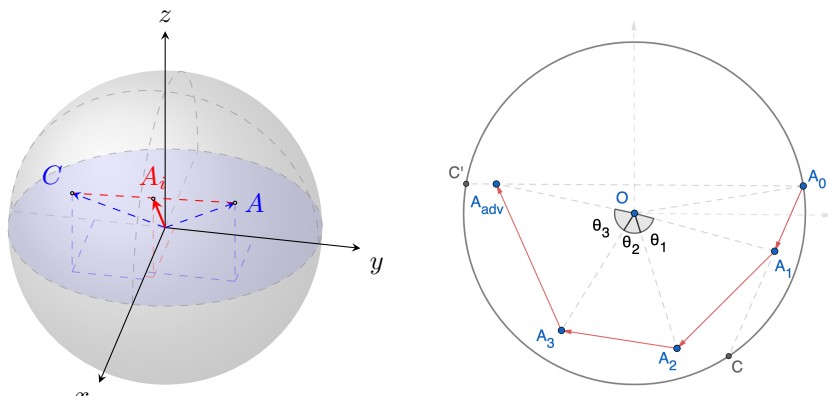

Figure 1: **L**: $A,C$ are latent image representations. $A_i$ lies on chord $AC$. Targeted latent space movements lower norm. **R**: Initial radial loss towards $O$ forms $A_1$ with implied movement towards $C$. $A_2, \ldots, A_{adv}$ follow. Implied movement is towards $C'$ - further than $C$ from $A$ in $\theta$ (angles).

Both losses depend on the layer $i$. With access till layer $D$, multiple candidate $i$ values exist. For simplicity, we only use $D, D-1$ and replace $\text{sign}(\nabla L_{\text{init}})$ with $\frac{1}{2}[\text{sign}(\nabla L_{\text{init},D-1}) + \text{sign}(\nabla L_{\text{init},D})]$ where the subscript $L_{\text{init},D-1}$ denotes setting $i$ as $D-1$, and similarly we set $i = D$ and average the sign for $\text{sign}(\nabla L_{\text{radial}})$. This averaged term becomes $\Psi$ in our pseudocode (Algorithm 1). We clamp $x^k$ iterates to satisfy $\|x^k - x^0\| \le \epsilon$. Our attack resembles ILAP, with the initial direction $\Delta_i^{adv}$ from an unsupervised radial loss without labels. We then maximize the angular deviation from the original latent, and do not "follow the leader" on the original angular deviation after $t_{init}$. The intuition is as follows. The surrogate loss $-\|z^t\|$ moves along a chord in the hypersphere's interior from $z^0$ towards $-z^0$, the most dissimilar point, while latent representations are on the surface . We use the last 2 layers for simplicity - stronger attacks might exist using all layers. Concentration of norm forms the hypersphere, and the 2-step process is key. 1-step methods, e.g. single-stage works discussed under related works, use methods such as random initialization and angular minimization, but perform worse (see Appendix P). The concentration is in $L_2$, but the adversarial example is in the $L_\infty$ metric.

---

**Algorithm 1** Angular attack algorithm

---

**Input:** Neural Network $N$; access of model parameters of $N$ till layer $D$, $N_{1,i}$: sub-network of $N$ upto layer $i$ , a starting real sample $x^0$, perturbation $\alpha$
**Result:** Adversarial example $x^{t_{\text{radial}}}$
$t_{\text{init}} \leftarrow$ Num iterations with loss $L_{\text{init}}$ ;    $t_{\text{radial}} \leftarrow$ Num iterations with loss $L_{\text{radial}}$
$k \leftarrow 1, n \leftarrow t_{\text{init}} + 1$
**while** $k \le t_{\text{init}}$ **do**
$\quad L_{\text{init},i}^{k-1} = -\|N_{1,i}(x^{k-1})\|$ ;    $\Psi = \text{sign}\left(\nabla L_{\text{init},D-1}^{k-1}\right) + \text{sign}\left(\nabla L_{\text{init},D}^{k-1}\right)$
$\quad x^k \leftarrow x^{k-1} + \frac{\alpha}{2} \times \Psi$;    $x^k \leftarrow \text{clamp}(x^k, \epsilon)$ ;    $k \leftarrow k+1$
**end while**
**while** $n \le t_{\text{radial}}$ **do**
$\quad L_{\text{radial},i}^{n-1} = -\dfrac{\langle N_{1,i}(x^0), N_{1,i}(x^{n-1}) \rangle}{\|N_{1,i}(x^0)\| \|N_{1,i}(x^{n-1})\|}$ ;    $\Psi = \text{sign}\left(\nabla L_{\text{radial},D-1}^{n-1}\right) + \text{sign}\left(\nabla L_{\text{radial},D}^{n-1}\right)$
$\quad x^n \leftarrow x^{n-1} + \frac{\alpha}{2} \times \Psi$;    $x^n \leftarrow \text{clamp}(x^n, \epsilon)$ ;    $n \leftarrow n+1$
**end while**
**Return:** $x^{t_{\text{radial}}}$

---

Moving from $z^0$ along $-\nabla\|z^t\|$ does **not** move linearly towards the origin when the optimization is imperfect - e.g. using the gradient sign. We manipulate $x^i$, not $z^i$ - it may not be $\exists\, x'$ such that $N_{1,i}(x') = \gamma z^0, \forall \gamma, 0 \le \gamma < 1$ (the ray joining origin-$z^0$ may lack points with pre-images in $x$). Another issue is $\alpha$ movement direction-wise (ignoring sign averaging) - e.g. let $Z_i \in \mathbb{R}^2$, $N_{1,i}(x) = x$ - the representation as identity map. Let $x = (1, \epsilon)$ lie on the hypersphere of radius $\sqrt{1+\epsilon^2}$. The sign of $\nabla L_{\text{init}}$ is $(-,-)$ and $x^0 = (1-\alpha, \epsilon - \alpha)$ - not necessarily collinear with $(0,0) - (1,\epsilon)$. Batch

normalization leaks mean/variance to adversaries. Suppose we had a perfect network $N$ (could find $x$ mapping to any $z$, i.e., was onto), and **were assured** the latent space was a zero mean hypersphere and we get adversarial instances if we move the latent from $z$ to $-z$. This would be insufficient if all latents were translated by some vector $c$, i.e., the mean of the hypersphere was nonzero. To find the point opposite $z$ on the new hypersphere ($c - (z - c)$), we need $c$. Batch norm gives us this $c$. This weakness outweighs knowing $z'$ arising from $x'$ of a different label, as we show in the results.

## 2.2   When Does the Norm Concentrate ?

We assumed that after a batch normalization step, $\|Z\|$ concentrates. Consider sufficiency conditions. Let $Z_{ij}$ be the $j$-th entry of the $i$-th latent $Z_i$ formed from $X_i$ with expectation 0, variance 1 (batch norm). Consider independent instances $X_i$ and assume no instances indexed by $i$ were used for training to impact model parameters, which affects the other latents and makes them dependent - i.e. consider test/validation sets. Then, $Z_{ij}, Z_{i'j'}$ are independent if $i \neq i', \forall j' = j$. Now :

$$\|Z_i\|^2 = \sum_{j=1}^{d} Z_{ij}^2$$

This sums $d$ random variables (assuming $Z_i$ has dimension $d$) and depends on independence structures and the marginal distributions of each variable. Under certain cases, independence of $Z_{ij}, Z_{ij'}$ can occur. Suppose $Z_{ij}$ was constrained in $\{-1, 1\}$, and had to encode $2^D$ $X_i$'s. Then, $Z_{ij}, Z_{ij'}$ are independent under the optimal encoding, assigning for each $X_i$ a unique $D$-length code with entries $\in \{-1, 1\}$. When $Z_{ij}, Z_{ij'}$ are independent, $\|Z_i\|^2 = \sum_j Z_{ij}^2$ will concentrate around its expected value of $d$ under mild conditions e.g. bounded 4-th moments of $Z_{ij}$ (Chebyshev's inequality), but concentration tails may not be sub-Gaussian. By tails of concentrations, we mean bounds of form:

$$P\left(\left|\|Z_i^2\| - E(\|Z_i^2\|)\right| \geq \delta\right) \leq f(\delta)$$

For example, sub-Gaussian concentration implies $f(\delta) = O(\exp(-\delta^2))$. Suppose each $Z_{ij}$ is marginally distributed as a (sub)-Gaussian. With independence, Chernoff's inequality leads to a norm concentration for $\|Z\|$ with stronger sub-Gaussian tails over Chebyshev's (tails of $\frac{1}{\delta^2}$). Inequalities and implied tails can be consulted from e.g. [Vershynin, 2018]. We consider some previous results. For deep linear models of width(dimensionality) $d$, we have [Daneshmand et al., 2020, 2021]:

$$E[D_{KL}(Z\|\mathcal{N}(0, I_d))] = O\left((1 - \alpha)^i + \frac{b}{\alpha\sqrt{d}}\right)$$

$Z$ is the distribution after $N_{1,i}$ i.e., $i$ layers, $\alpha$ is a constant, $b$ is the training batch size. The LHS (termed the orthogonality gap) indicates that the distribution of the latents convergences to the isotropic Gaussian using KL divergence, which is a stronger condition leading to the latent norms being clustered around $\sqrt{d}$. **Though derived for linear networks, the original paper offers evidence that the conjecture holds for general multi-layer networks.** This indicates deeper layers i.e., increasing $i$ will have the largest effect on our assumptions holding, as this exponentially drops the KL divergence to a Gaussian (and concentrates the norm), with a secondary benefit from increasing width ($d$). Beyond linear networks, general networks with depth and width reduce to **Gaussian Processes** [Yang et al., 2019, Yang and Schoenholz, 2018, Yang and Hu, 2020, Yang, 2019, Neal, 2012]. These results usually apply to marginal distributions and not independence structures. Empirically, we find dependence between different variables (measured as correlation) falls as the network latent grows deeper, which aligns with theory. Dimensionality falls as the network deepens - recall the toy example of encoding $2^D$ instances with $D$ binary variables which forces independence. ILAP's empirical ablation studies found the optimal $i$ to create the latents for the attack occurred in the range 0.6 to 0.8 (network depth normalized to 1). In sum, we expect our **optimal layers for the latent to lie near the end of the network**. Our methods work without assumptions in hyperspherical representation spaces [Wang and Isola, 2020, Schroff et al., 2015, Mettes et al., 2019]. Analysis and empirical findings on the concentration and tails is included in Appendix R.

## 3   Results

We carry out an extensive set of experiments on Imagenet [Russakovsky et al., 2015], utilizing several ResNet [He et al., 2016] models - ResNet-$\{18, 34, 50, 101, 152\}$ and EfficientNet models [Tan and

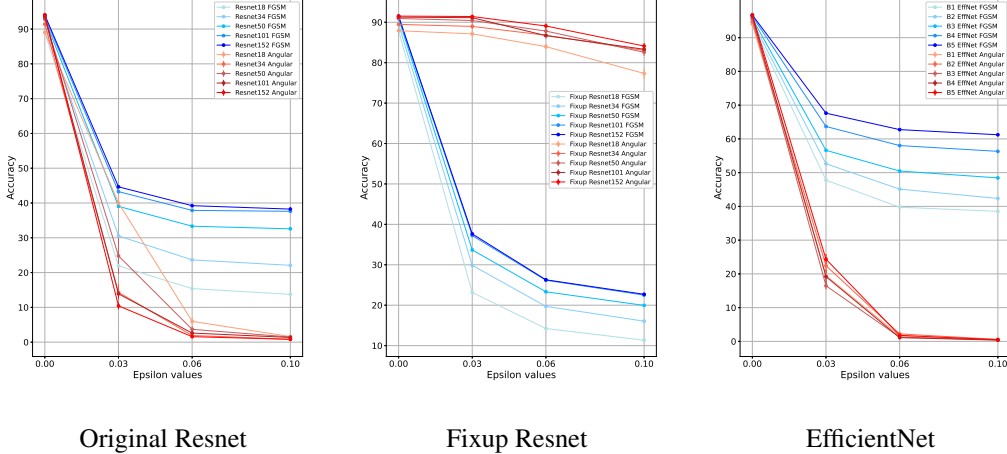

| Original Resnet | Fixup Resnet | EfficientNet |

Figure 2: Accuracies (%) of networks under FGSM (blue) and our Angular attack (red). (Color intensity proportional to network size). Accuracies under angular attack are lower without Fixup.

Le, 2019] B-1 to B-5. We chose these models because they have a common structure at construction. Except a few initial layers, they are constructed by repeatedly stacking the same blocks ("Basic" or "Bottleneck" blocks for ResNets, and Mobile Convolutional blocks [Howard et al., 2017] for EfficientNets). Every block possesses at least one batch norm layer, allowing the extraction of the appropriate latent. All models were ran on Imagenet with a standard normalization pre-processing step. Results of all attacks on all models, a sanity check with a fixed norm model (explicitly hyperspherical, i.e. always concentrated in norm), improving supervised (label present) cases on top of FGSM/PGD, comparisons to single-stage attacks previously proposed, interactions with defence, ablations, visualizations, statistical testing, confidence intervals and other transfer learning results and comparisons appear in the Appendices. As the alternative to batchnormed models, we choose Fixup [Zhang et al., 2019] - a different way to initialize the models. Core results are summarized in Figure 2. On the batchnorm-free Fixup Resnet, the FGSM attack greatly outperforms our angular attack. On the other two architectures, this is **not** the case. FGSM's performance and clean accuracy vary far less than the success of the angular attack. That implies overall model quality and robustness to common attacks remains the same, and vulnerability to the angular attack specifically is what varies. We train Fixup resnets on imagenet as alternatives to the batchnormed models for Resnet only, as methods in Fixup do not generalize to efficientnets. Details of these training steps are in the Appendix B, and we also attach our training and inference codebases. We obtained the models from publicly available repositories for PyTorch [Paszke et al., 2019], namely torchvision and EfficientNets-PyTorch, and did not change the original weights. For the choice of $\epsilon$ for the adversarial attack, we chose $0.03, 0.06, 0.1$ and carried out all attacks using an $\alpha = 0.01$ over $40$ iterations.

**Vision Transformers and LayerNorm.** Recently, Layer normalization [Ba et al., 2016] has emerged as an alternate mode of normalization to batch norm. It finds prominent usage in the transformer architecture [Vaswani et al., 2017] and for images, in the ViT (Vision Transformer) models. We used our attacks on the case of ViT models [Dosovitskiy et al., 2020] (B/L)-(16/32). In all cases, the attacks were successful as with Resnets/batchnorm. This indicates our attack can succeed even when instead of batch normalization, layer normalization is utilized instead. We discuss this in Appendix A, but make batch norm the focus as it has more pre-existing theoretical analysis [Daneshmand et al., 2020].

We examine the results on all Resnets in Tables 1,2,3 with Table 8 denoting clean accuracies. Angular attacks - especially in terms of their top 5 accuracy, at higher $\epsilon$ and on deeper networks - dominate FGSM attacks, while losing to PGD attacks. No random perturbation performs well. Confidence intervals rule out a fluke (statistical tests Appendix C). Raw numbers for Fixup Resnets (in Appendix D) exhibit a different pattern - the angular attack performs as well as a random method, while FGSM, and PGD perform as normal. EfficientNets (in Appendix E) follow ResNets in their result. In Table 4, we exhibit the max over single-stage benchmarks among [Zhou et al., 2018, Inkawhich et al., 2019,

Table 1: Comparison of FGSM and Random noise, Top-1 accuracy, Top-5 in brackets.

| Net Type | $\epsilon = 0.03$ | | $\epsilon = 0.06$ | | $\epsilon = 0.1$ | |
|---|---|---|---|---|---|---|
| | FGSM | Random | FGSM | Random | FGSM | Random |
| Resnet-18 | 1.95 (21.93) | 69.40 (88.94) | 1.16 (15.39) | 68.58 (88.51) | 1.22 (13.74) | 67.11 (87.46) |
| Resnet-34 | 4.36 (30.54) | 73.06 (91.31) | 2.95 (23.65) | 72.43 (90.96) | 3.13 (22.08) | 71.23 (90.26) |
| Resnet-50 | 8.18 (39.04) | 75.79 (92.07) | 6.62 (33.33) | 74.89 (92.32) | 7.10 (32.58) | 73.57 (91.58) |
| Resnet-101 | 9.93 (43.27) | 77.24 (93.52) | 8.32 (37.88) | 76.79 (93.25) | 9.10 (37.61) | 75.57 (92.75) |
| Resnet-152 | 10.23 (44.65) | 77.21 (93.78) | 8.76 (39.23) | 76.76 (93.48) | 9.82 (38.25) | 76.28 (92.98) |

Table 2: Comparing Angular attacks, top-1/5 accuracy, with confidence intervals on different Resnets.

| Type | $\epsilon = 0.03$ | | $\epsilon = 0.06$ | | $\epsilon = 0.1$ | |
|---|---|---|---|---|---|---|
| | Top-1 | Top-5 | Top-1 | Top-5 | Top-1 | Top-5 |
| 18 | 23.416 ±1.22 | 40.034 ±2.12 | 2.422 ±0.52 | 5.980 ±1.12 | 0.454 ±0.11 | 1.602 ±0.39 |
| 34 | 7.164 ±1.28 | 14.320 ±2.48 | 0.592 ±0.64 | 1.916 ±0.32 | 0.216 ±0.04 | 0.814 ±0.27 |
| 50 | 13.968 ±2.82 | 24.796 ±5.22 | 1.418 ±0.22 | 3.680 ±0.83 | 0.432 ±0.12 | 1.478 ±0.26 |
| 101 | 7.012 ±0.62 | 13.886 ±1.23 | 0.874 ±0.126 | 2.606 ±0.78 | 0.390 ±0.122 | 1.346 ±0.28 |
| 152 | 5.030 ±0.4 | 10.438 ±1.02 | 0.534 ±0.12 | 1.572 ±0.265 | 0.248 ±0.05 | 0.808 ±0.21 |

Table 3: Comparing PGD attacks, top-1/5 accuracy, with confidence intervals on Resnets.

| Type | $\epsilon = 0.03$ | | $\epsilon = 0.06$ | | $\epsilon = 0.1$ | |
|---|---|---|---|---|---|---|
| | Top-1 | Top-5 | Top-1 | Top-5 | Top-1 | Top-5 |
| 18 | 0.006 ±0.001 | 3.374 ±0.04 | 0.0 ±0.0 | 0.484 ±0.04 | 0.0 ±0.0 | 0.17 ±0.02 |
| 34 | 0.006 ±0.001 | 4.386 ±0.03 | 0.006 ±0.001 | 0.666 ±0.04 | 0.004 ±0.001 | 0.288 ±0.03 |
| 50 | 0.028 ±0.007 | 8.076 ±1.23 | 0.006 ±0.002 | 2.914 ±0.4 | 0.0 ±0.0 | 1.956 ±0.3 |
| 101 | 0.032 ±0.07 | 9.648 ±1.76 | 0.008 ±0.002 | 3.902 ±0.62 | 0.006 ±0.002 | 2.702 ±0.54 |
| 152 | 0.042 ±0.08 | 9.864 ±1.83 | 0.014 ±0.003 | 4.16 ±0.73 | 0.015 ±0.004 | 3.253 ±0.64 |

Table 4: Comparing single-stage attacks, top-1/5 accuracy, with confidence intervals on Resnets.

| Type | $\epsilon = 0.03$ | | $\epsilon = 0.06$ | | $\epsilon = 0.1$ | |
|---|---|---|---|---|---|---|
| | Top-1 | Top-5 | Top-1 | Top-5 | Top-1 | Top-5 |
| 18 | 30.45 | 48.34 | 6.98 | 16.87 | 3.89 | 10.22 |
| 34 | 16.78 | 35.89 | 1.45 | 8.32 | 1.68 | 4.54 |
| 50 | 22.68 | 39.10 | 4.62 | 10.45 | 4.71 | 6.58 |
| 101 | 15.65 | 34.67 | 5.82 | 11.28 | 5.28 | 8.95 |
| 152 | 8.29 | 18.53 | 1.78 | 5.67 | 0.95 | 3.22 |

Table 5: Top-1 (& Top-5) accuracy for Vision Transformers: $\epsilon = 0.1$

| Model | FGSM | PGD | Angular |
|---|---|---|---|
| ViT-B-16 | 35.1 (60.6) | 5.7 (18.5) | 16.7 (28.5) |
| ViT-B-32 | 34.8 (59.4) | 5.4 (21.1) | 17.8 (31.8) |
| ViT-L-16 | 36.0 (65.6) | 8.2 (17.7) | 14.5 (33.8) |
| ViT-L-32 | 34.1 (61.7) | 5.8 (22.9) | 13.9 (38.5) |

Table 6: Top-1/5 accuracy for ResNets, angular attack not on BN layer, $\epsilon = 0.03$

| ResNet | Top-1 | Top-5 |
|---|---|---|
| 18 | 27.3 | 46.2 |
| 34 | 11.5 | 19.2 |
| 50 | 17.3 | 28.5 |
| 101 | 11.32 | 19.5 |
| 152 | 8.2 | 11.9 |

Table 7: Resnet-34 ablations. Last 2 layers of the net granted access to are used for angular attack.

| Access till | $\epsilon = 0.03$ | | $\epsilon = 0.06$ | | $\epsilon = 0.1$ | |
|---|---|---|---|---|---|---|
| | Top 1 | Top 5 | Top 1 | Top 5 | Top 1 | Top 5 |
| 8 | 54.46 ±1.78 | 77.35 ±0.68 | 18.44 ±0.45 | 32.68 ±0.68 | 4.95 ±0.86 | 11.68 ±1.26 |
| 9 | 35.84 ±0.78 | 59.32 ±0.82 | 6.32 ±0.56 | 13.84 ±0.56 | 1.39 ±0.37 | 13.84 ±1.78 |
| 10 | 26.22 ±1.56 | 41.5 ±1.45 | 1.58 ±0.38 | 3.88 ±1.02 | 0.60 ±0.12 | 1.52 ±0.37 |
| 11 | 13.20 ±0.26 | 22.68 ±0.86 | 0.94 ±0.31 | 2.34 ±0.22 | 0.34 ±0.06 | 0.92 ±0.17 |
| 12 | 6.40 ±0.72 | 12.05 ±1.89 | 0.45 ±0.12 | 2.27 ±0.56 | 0.20 ±0.08 | 0.65 ±0.22 |

Table 8: Clean accuracies for Resnets

| ResNet type | 18 | 34 | 50 | 101 | 152 |
|---|---|---|---|---|---|
| Acc@1 | 69.75 | 73.31 | 76.13 | 77.37 | 78.31 |
| Acc@5 | 89.07 | 91.42 | 92.86 | 93.54 | 94.04 |

Table 9: Resnet-18 absolute correlations

| Block group | 1 | 2 | 3 |
|---|---|---|---|
| Absolute mean correlation | 0.41 | 0.32 | 0.19 |

Table 10: Angular transfer attack results and targeted benchmarks

a) Angular results on CIFAR-100, Resnet-50

| Method | $\epsilon = 0.03$ | | $\epsilon = 0.06$ | | $\epsilon = 0.1$ | |
|---|---|---|---|---|---|---|
| | Top 1 | Top 5 | Top 1 | Top 5 | Top 1 | Top 5 |
| PGD | 0.16 | 0.20 | 0.14 | 0.14 | 0.14 | 0.14 |
| Random | 64.42 | 88.18 | 53.1 | 79.76 | 38.63 | 66.23 |
| FGSM | 29.28 | 59.02 | 25.86 | 51.95 | 15.01 | 35.14 |
| Angular | 2.32 | 10.79 | 2.16 | 9.72 | 1.88 | 9.49 |

b) Targeted benchmark on Resnets

| Net Type | $\epsilon = 0.03$ | | $\epsilon = 0.06$ | |
|---|---|---|---|---|
| | Top 1 | Top 5 | Top 1 | Top 5 |
| Resnet-18 | 25.16 | 42.58 | 1.46 | 5.85 |
| Resnet-34 | 12.2 | 21.8 | 1.9 | 2.86 |
| Resnet-50 | 13.55 | 30.97 | 0.65 | 5.16 |
| Resnet-101 | 5.85 | 13.66 | 2.86 | 6.67 |
| Resnet-152 | 9.68 | 16.77 | 1.29 | 3.87 |

Ganeshan et al., 2019, Inkawhich et al., 2020b, Zhang et al., 2022a, Wang et al., 2021, Inkawhich et al., 2020a]. The performance is worse than our method. We also showcase results of ViT models on $\epsilon = 0.1$ for imagenet in Table 5. (Results for other values of $\epsilon$ are in Appendix F). Finally in Table 6, we attack Resnets with the angular attack but move the layer being attacked to the layer before the BN layer. This noticeably weakens the attack, suggesting a relationship between the two.

We examine the drop in accuracy as a function of the access to various blocks. We run our attacks consistently accessing the $3/4$-th layer of a net, i.e., for a Resnet-50 of blocks $[3, 4, 6, 3]$ - total 16 - we use only the first 12 blocks and sum our angular losses over block $11, 12$. On Resnet-34 access to deeper blocks strongly strengthens the attack in Table 7. This agrees with the theory of batch normalization discussed previously [Daneshmand et al., 2021] and with the empirical findings of ILAP [Huang et al., 2019], which found that the most optimal layers to perturb lay between $50\%$ and $80\%$ of the network's length. Further depth ablations are in Appendix G. Concentration of norm relies on convergence of batchnorm and independence. Assuming the former, we can check independence by examining the absolute, off-diagonal correlations in the correlation matrices of the latent representations. We note a decorrelation effect with depth, which matches the success of our attacks. We compare the fall in absolute correlation among the latent dimensions in Table 9, across three "block groups" of Resnet-18 which has 8 blocks organized as $[2, 2, 2, 2]$. Correlation - a proxy for independence - decreases over the last three block groups.

**Transfer learning.** For transfer learning, we add a linear classifier that can access the last extracted latent layer of the resnet, and we unfreeze this classifier and the last group of blocks. Every other layer is frozen. The setup is fine-tuned, downstream, on CIFAR-10 and CIFAR-100 [Krizhevsky et al., 2009]. We show results on CIFAR-100 with Resnet-50 in Table 10. Although Table 10 indicates that targeted attacks perform worse than the original angular attacks, it does not mean that we cannot use the label information. It only means that when we randomly pick an instance of a different label, the angular attack often does better. But, the combination of label and angular attack can do better than either attack, so having an instance of a different label can still be very helpful. We provide some cases where our loss is provided in ensemble with the targeted loss and these perform better than either (see Appendix H). Results for CIFAR-10 / other resnets/datasets are in Appendix J. Our method outperforms all but PGD.

**Ablation against other cases.** The radial loss is an implicit movement towards the opposite point on a hypersphere. This assumes that the hypersphere exists at all. If we have a different latent $Z_j$ from a point $X_j$ of a different label from $X_i$, we check if our attack might be more effective using an initial loss of the form $-\|Z_i - Z_j\|$. Increasing this loss would be lowering the distance between $Z_i, Z_j$ i.e. moving towards the other point in latent space. Yet, this **targeted attack** - shown in Table 10 - is actually inferior to our attack, which lacks the access.

## 4 Conclusion

We have shown a powerful, label-free attack which only utilizes a portion of all the layers available for a network to construct adversarial examples that fool the entire network. It succeeds at this without knowing the label or having gradient access to the full model, and these adversarial methods

generalize to the case where the model was fine-tuned afterwards. These results have relevance at the intersection of the theory and practice of adversarial robustness and the growing study of batch normalization and its weaknesses and drawbacks. We provide support to the notion that batch normalization may open up unseen avenues of perturbations that unexpectedly impact model robustness, a viewpoint supported by previous literature [Galloway et al., 2019, Benz et al., 2020, Wang et al., 2022]. We also extend our results to LayerNorm, which is increasingly relevant with the advent of transformer architectures [Dosovitskiy et al., 2020].

## Acknowledgments and Disclosure of Funding

Resources used in this work were provided by the Province of Ontario, the Government of Canada through CIFAR, companies sponsoring the Vector Institute `https://vectorinstitute.ai/partners/` and the Natural Sciences and Engineering Council of Canada.

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

## A  Layer normalization

Layer normalization [Ba et al., 2016] is similar to batch normalization. However, unlike batch norm, Layer norm computes the mean and standard deviation over all activations in a layer. This further means it can be used in the online setting or in the test setting without caching statistics, as the statistics are calculated in real time.

Now, it is clear that if the input is of the form $(N, H)$ where $N$ denotes the minibatch size and $H$ the hidden dimension, layer normalization ensures that each instance $i$ satisfies, for a layer of dimension $d$ over all activations $H_{i,j}$ :

$$\sum_{1 \leq j \leq d} H_{i,j} = 0, \ \sum_{1 \leq j \leq d} H_{i,j}^2 = d$$

i.e. a hyperspherical space. This means our assumptions hold exactly. However, this assumes the layer normalization occurs over the entire hidden dimension. This may not necessarily be the case. For example, if the dimensions are $(N, C, H, W)$ it is possible that normalization only occurs over one of the dimensions of $C, H, W$. However, in this case, we can still recover an appropriate hyperspherical latent by summing over the remaining dimensions.

## B  Training Details

Recall that we use the losses of the following form (section 2.1 of the main text) :

$$L_{\text{radial}} = -\frac{\langle z_i^0, z_i^t \rangle}{\|z_i^0\|\|z_i^t\|}; z_i^0 = N_{1,i}(x^0), z_i^t = N_{1,i}(x^t)$$

$$L_{\text{init}}^t = -\|z^t\| = -\|N_{1,i}(x^t)\|$$

We take the last two layers upto which we have access as $i, i'$ to sum the signs of the angular loss. We cut the $\alpha$ by 2 to adjust for the sum of the signs, resulting in the overall averaging of signs referenced in the text. Also, before switching over to the angular loss, we can keep using $L_{init}$ for more than the initial iterations. We utilize 20 such iterations before switching, at $\alpha' = \alpha/40$. Hence, in total, our **angular PGD attack** consists of 20 iterations of finding an unsupervised radial direction of moving to the antipodal point in latent space, and 20 iterations of maximizing the angular deviation given the initial movement.

Projection back to the valid PyTorch Tensor space for the adversarial instance (adjusting for normalization layers) proceeds as per normal PGD methods. The $\epsilon$ values here are calculated in the normalized Tensor space i.e. after normalizing the $[0, 1]$ tensor Imagenet image with parameters of mean as $[0.485, 0.456, 0.406]$ and standard deviation as $[0.229, 0.224, 0.225]$ channelwise. The projection is such that it respects both the original tensor's range and the $\epsilon$ in the normalized space.

Normalization settings for Imagenet for Resnet were kept as-is from the pytorch examples [2] and also as per the Fixup repository[3] as well as for Efficientnet[4]. For training, we utilize SGD. Please consult the attached codebase for all code-based details. Everything was run on a single Tesla V100 GPU, on Torch 1.6, torchvision 0.7.0. All confidence intervals appearing in this paper were generated by bootstrapping and asymmetric intervals were written in $\pm$ of the max one sided deviation. We use the Wilcoxon sign-rank test for determination of statistical significance, with $p < 10^{-3}$ as the cutoff, we refer to anything more significant than this $p$-value as passing the test here on. Cosine learning rate is used to train the Fixup resnet as per [Zhang et al., 2019].

---

[2]https://github.com/pytorch/examples/blob/main/imagenet/main.py
[3]https://github.com/hongyi-zhang/Fixup
[4]https://github.com/lukemelas/EfficientNet-PyTorch/tree/master/examples/imagenet

# C Original ResNets

We recall that in this class fall 5 models, namely Resnet-18,34,50,101, and 152. In terms of block count, they respectively possess $8, 16, 16, 33, 50$ layers. All radial attacks are performed with the last 2 layers upto which we have access. Access is granted upto blocks $6, 12, 12, 25, 38$ respectively. So for example, on Resnet-18 the radial attack uses the loss signal from blocks $5, 6$ alone to craft the example. **Every value provided in every table is a percentage accuracy metric.**

Table 11: Clean accuracies for ResNets

| ResNet type | 18 | 34 | 50 | 101 | 152 |
|---|---|---|---|---|---|
| Acc@1 | 69.75 | 73.31 | 76.13 | 77.37 | 78.31 |
| Acc@5 | 89.07 | 91.42 | 92.86 | 93.54 | 94.04 |

Table 12: Comparison of FGSM and Random noise, Top-1 accuracy.

| Net Type | $\epsilon = 0.03$ | | $\epsilon = 0.06$ | | $\epsilon = 0.1$ | |
|---|---|---|---|---|---|---|
| | FGSM | Random | FGSM | Random | FGSM | Random |
| Resnet-18 | 1.95 | 69.40 | 1.16 | 68.58 | 1.22 | 67.11 |
| Resnet-34 | 4.36 | 73.06 | 2.95 | 72.43 | 3.13 | 71.23 |
| Resnet-50 | 8.18 | 75.79 | 6.62 | 74.89 | 7.10 | 73.57 |
| Resnet-101 | 9.93 | 77.24 | 8.32 | 76.79 | 9.10 | 75.57 |
| Resnet-152 | 10.23 | 77.21 | 8.76 | 76.76 | 9.82 | 76.28 |

Table 13: Comparison of FGSM and Random noise, Top-5 accuracy.

| Net Type | $\epsilon = 0.03$ | | $\epsilon = 0.06$ | | $\epsilon = 0.1$ | |
|---|---|---|---|---|---|---|
| | FGSM | Random | FGSM | Random | FGSM | Random |
| Resnet-18 | 21.93 | 88.94 | 15.39 | 88.51 | 13.74 | 87.46 |
| Resnet-34 | 30.54 | 91.31 | 23.65 | 90.96 | 22.08 | 90.26 |
| Resnet-50 | 39.04 | 92.07 | 33.33 | 92.32 | 32.58 | 91.58 |
| Resnet-101 | 43.27 | 93.52 | 37.88 | 93.25 | 37.61 | 92.75 |
| Resnet-152 | 44.65 | 93.78 | 39.23 | 93.48 | 38.25 | 92.98 |

Table 14: Comparison of PGD attacks, top-1 and top-5 accuracy, with confidence intervals on different Resnet types.

| Type | $\epsilon = 0.03$ | | $\epsilon = 0.06$ | | $\epsilon = 0.1$ | |
|---|---|---|---|---|---|---|
| | Top-1 | Top-5 | Top-1 | Top-5 | Top-1 | Top-5 |
| 18 | $0.006 \pm 0.001$ | $3.374 \pm 0.04$ | $0.0 \pm 0.0$ | $0.484 \pm 0.04$ | $0.0 \pm 0.0$ | $0.17 \pm 0.02$ |
| 34 | $0.006 \pm 0.001$ | $4.386 \pm 0.03$ | $0.006 \pm 0.001$ | $0.666 \pm 0.04$ | $0.004 \pm 0.001$ | $0.288 \pm 0.03$ |
| 50 | $0.028 \pm 0.007$ | $8.076 \pm 1.23$ | $0.006 \pm 0.002$ | $2.914 \pm 0.4$ | $0.0 \pm 0.0$ | $1.956 \pm 0.3$ |
| 101 | $0.032 \pm 0.07$ | $9.648 \pm 1.76$ | $0.008 \pm 0.002$ | $3.902 \pm 0.62$ | $0.006 \pm 0.002$ | $2.702 \pm 0.54$ |
| 152 | $0.042 \pm 0.08$ | $9.864 \pm 1.83$ | $0.014 \pm 0.003$ | $4.16 \pm 0.73$ | $0.015 \pm 0.004$ | $3.253 \pm 0.64$ |

Table 15: Comparison of Angular attacks, top-1 and top-5 accuracy, with confidence intervals on different Resnet types.

| Type | $\epsilon = 0.03$ | | $\epsilon = 0.06$ | | $\epsilon = 0.1$ | |
|------|-------|-------|-------|-------|-------|-------|
|  | Top-1 | Top-5 | Top-1 | Top-5 | Top-1 | Top-5 |
| 18 | 23.416 ±1.22 | 40.034 ±2.12 | 2.422 ±0.52 | 5.980 ±1.12 | 0.454 ±0.11 | 1.602 ±0.39 |
| 34 | 7.164 ±1.28 | 14.320 ±2.48 | 0.592 ±0.64 | 1.916 ±0.32 | 0.216 ±0.04 | 0.814 ±0.27 |
| 50 | 13.968 ±2.82 | 24.796 ±5.22 | 1.418 ±0.22 | 3.680 ±0.83 | 0.432 ±0.12 | 1.478 ±0.26 |
| 101 | 7.012 ±0.62 | 13.886 ±1.23 | 0.874 ±0.126 | 2.606 ±0.78 | 0.390 ±0.122 | 1.346 ±0.28 |
| 152 | 5.030 ±0.4 | 10.438 ±1.02 | 0.534 ±0.12 | 1.572 ±0.265 | 0.248 ±0.05 | 0.808 ±0.21 |

At higher values of $\epsilon$, angular attack outperforms the FGSM one. The confidence intervals (constructed by bootstrapping, denoted by ±) mark the 5 and 95 percentile confidence intervals and can be used to gauge statistical significance. The wilcoxon signed rank test passed for every case where the point estimate of the angular attack was better than the FGSM one, with the exception of top-1 on Resnet-101.

## D   Fixup ResNets

Just as with original Resnets, here we have 5 models, namely Resnet-18,34,50,101, and 152. In terms of block count, they respectively possess $8, 16, 16, 33, 50$ layers. All radial attacks are performed with the last 2 layers upto which we have access. Access is granted upto blocks $6, 12, 12, 25, 38$ respectively. So for example, on FixUpResnet-18 the radial attack uses the loss signal from blocks $5, 6$ alone to craft the example. **Every value provided in every table is a percentage accuracy metric.** In short, everything is performed just as with the original Resnets.

Table 16: Clean accuracies for ResNets

| ResNet type | 18 | 34 | 50 | 101 | 152 |
|---|---|---|---|---|---|
| Acc@1 | 68.212 | 70.466 | 72.938 | 73.596 | 73.866 |
| Acc@5 | 87.910 | 89.470 | 90.932 | 91.273 | 91.528 |

Table 17: Comparison of FGSM and Random noise, Top-1 accuracy.

| Net Type | $\epsilon = 0.03$ | | $\epsilon = 0.06$ | | $\epsilon = 0.1$ | |
|---|---|---|---|---|---|---|
| | FGSM | Random | FGSM | Random | FGSM | Random |
| Resnet-18 | 2.09 | 68.18 | 1.25 | 67.61 | 1.19 | 66.28 |
| Resnet-34 | 3.73 | 70.79 | 2.18 | 70.39 | 2.08 | 69.36 |
| Resnet-50 | 5.34 | 73.29 | 3.63 | 72.80 | 3.51 | 71.61 |
| Resnet-101 | 6.37 | 74.41 | 4.17 | 73.93 | 4.17 | 72.76 |
| Resnet-152 | 6.59 | 74.68 | 4.71 | 74.21 | 4.60 | 73.06 |

Table 18: Comparison of FGSM and Random noise, Top-5 accuracy.

| Net Type | $\epsilon = 0.03$ | | $\epsilon = 0.06$ | | $\epsilon = 0.1$ | |
|---|---|---|---|---|---|---|
| | FGSM | Random | FGSM | Random | FGSM | Random |
| Resnet-18 | 23.17 | 87.91 | 14.23 | 87.56 | 11.33 | 86.75 |
| Resnet-34 | 29.86 | 89.52 | 19.71 | 89.28 | 16.05 | 88.58 |
| Resnet-50 | 33.67 | 91.12 | 23.32 | 90.83 | 19.95 | 90.10 |
| Resnet-101 | 37.22 | 91.83 | 26.18 | 91.63 | 22.50 | 91.00 |
| Resnet-152 | 37.64 | 92.09 | 26.29 | 91.80 | 22.70 | 91.24 |

Table 19: Comparison of PGD attacks, top-1 and top-5 accuracy, with confidence intervals on different Resnet types.

| Type | $\epsilon = 0.03$ | | $\epsilon = 0.06$ | | $\epsilon = 0.1$ | |
|---|---|---|---|---|---|---|
| | Top-1 | Top-5 | Top-1 | Top-5 | Top-1 | Top-5 |
| 18 | 0.034 ±0.003 | 4.490 ±0.32 | 0.014 ±0.003 | 0.322 ±0.05 | 0.008 ±0.002 | 0.058 ±0.013 |
| 34 | 0.106 ±0.02 | 6.524 ±0.8 | 0.046 ±0.008 | 0.608 ±0.108 | 0.030 ±0.007 | 0.142 ±0.03 |
| 50 | 0.090 ±0.015 | 5.592 ±1.26 | 0.016 ±0.003 | 0.266 ±0.05 | 0.006 ±0.002 | 0.022 ±0.005 |
| 101 | 0.140 ±0.03 | 6.988 ±1.13 | 0.018 ±0.005 | 0.317 ±0.04 | 0.006 ±0.001 | 0.048 ±0.004 |
| 152 | 0.124 ±0.08 | 7.474 ±1.18 | 0.020 ±0.003 | 0.340 ±0.06 | 0.004 ±0.001 | 0.030 ±0.005 |

Table 20: Comparison of Angular attacks, top-1 and top-5 accuracy, with confidence intervals on different Resnet types.

| Type | $\epsilon = 0.03$ | | $\epsilon = 0.06$ | | $\epsilon = 0.1$ | |
|---|---|---|---|---|---|---|
| | Top-1 | Top-5 | Top-1 | Top-5 | Top-1 | Top-5 |
| 18 | 66.972 ±1.78 | 87.128 ±0.68 | 62.856 ±0.78 | 83.970 ±0.68 | 54.666 ±1.24 | 77.336 ±1.27 |
| 34 | 69.968 ±0.84 | 88.972 ±0.67 | 67.012 ±1.27 | 86.780 ±0.84 | 61.868 ±0.94 | 83.034 ±0.53 |
| 50 | 72.914 ±0.42 | 90.434 ±0.62 | 68.368 ±1.2 | 87.818 ±0.67 | 61.664 ±1.2 | 82.570 ±0.87 |
| 101 | 73.306 ±1.07 | 91.128 ±0.48 | 69.424 ±0.47 | 88.686 ±0.34 | 62.550 ±0.81 | 83.318 ±0.56 |
| 152 | 73.626 ±0.72 | 91.412 ±0.37 | 70.224 ±0.84 | 89.078 ±0.64 | 63.700 ±0.84 | 84.116 ±0.74 |

Unlike the original resnets, the angular values do not even come close to the FGSM counterparts. This strongly suggests that removal of batch norm completely fixes this mode of vulnerability.

# E EfficientNets

Here we have 5 models, namely EfficientNets B1 to B5. In terms of block count, they respectively possess $23, 23, 26, 32, 39$ layers. All radial attacks are performed with the last 2 layers upto which we have access. Access is granted upto blocks $17, 17, 19, 24, 31$ respectively. So for example, on B1 the radial attack uses the loss signal from blocks $16, 17$ alone to craft the example. **Every value provided in every table is a percentage accuracy metric.** In short, everything is performed just as with the original Resnets.

Table 21: Clean accuracies for EfficientNets

| Efficientnet type | B1 | B2 | B3 | B4 | B5 |
|---|---|---|---|---|---|
| Clean accuracy (top-1) | 78.382 | 79.808 | 81.532 | 83.026 | 83.778 |
| Clean accuracy (top-5) | 94.036 | 94.732 | 95.646 | 96.342 | 96.710 |

Table 22: Comparison of FGSM and Random noise, Top-1 accuracy.

| Net Type | $\epsilon = 0.03$ | | $\epsilon = 0.06$ | | $\epsilon = 0.1$ | |
|---|---|---|---|---|---|---|
| | FGSM | Random | FGSM | Random | FGSM | Random |
| B1 | 21.314 | 77.863 | 16.860 | 77.950 | 14.220 | 77.296 |
| B2 | 26.522 | 79.604 | 21.926 | 79.302 | 20.218 | 78.564 |
| B3 | 30.700 | 81.430 | 26.526 | 81.064 | 25.164 | 80.678 |
| B4 | 39.380 | 82.842 | 34.820 | 82.712 | 33.274 | 82.344 |
| B5 | 42.012 | 83.744 | 38.002 | 83.628 | 36.764 | 83.424 |

Table 23: Comparison of FGSM and Random noise, Top-5 accuracy.

| Net Type | $\epsilon = 0.03$ | | $\epsilon = 0.06$ | | $\epsilon = 0.1$ | |
|---|---|---|---|---|---|---|
| | FGSM | Random | FGSM | Random | FGSM | Random |
| B1 | 47.767 | 93.663 | 39.758 | 93.681 | 38.54 | 93.582 |
| B2 | 52.634 | 94.696 | 45.114 | 94.532 | 42.356 | 94.222 |
| B3 | 56.612 | 95.650 | 50.478 | 95.484 | 48.448 | 95.282 |
| B4 | 63.678 | 96.314 | 58.024 | 96.226 | 56.314 | 96.116 |
| B5 | 67.620 | 96.740 | 62.750 | 96.708 | 61.226 | 96.626 |

Table 24: Comparison of PGD attacks, top-1 and top-5 accuracy, with confidence intervals.

| Net Type | $\epsilon = 0.03$ | | $\epsilon = 0.06$ | | $\epsilon = 0.1$ | |
|---|---|---|---|---|---|---|
| | Top-1 | Top-5 | Top-1 | Top-5 | Top-1 | Top-5 |
| B1 | 0.430 ±0.082 | 3.318 ±0.43 | 0.078 ±0.012 | 0.386 ±0.05 | 0.017 ±0.004 | 0.197 ±0.022 |
| B2 | 0.618 ±0.102 | 3.092 ±0.52 | 0.064 ±0.006 | 0.340 ±0.07 | 0.012 ±0.002 | 0.218 ±0.04 |
| B3 | 0.698 ±0.108 | 2.482 ±0.56 | 0.136 ±0.02 | 0.484 ±0.07 | 0.044 ±0.006 | 0.318 ±0.07 |
| B4 | 0.590 ±0.08 | 2.112 ±0.4 | 0.054 ±0.009 | 0.470 ±0.08 | 0.018 ±0.004 | 0.380 ±0.05 |
| B5 | 0.812 ±0.12 | 1.728 ±0.35 | 0.135 ±0.02 | 0.260 ±0.04 | 0.073 ±0.008 | 0.125 ±0.03 |

Table 25: Comparison of Angular attacks, top-1 and top-5 accuracy, with confidence intervals.

| Net Type | $\epsilon = 0.03$ | | $\epsilon = 0.06$ | | $\epsilon = 0.1$ | |
|---|---|---|---|---|---|---|
| | Top-1 | Top-5 | Top-1 | Top-5 | Top-1 | Top-5 |
| B1 | 11.479 ±1.25 | 19.501 ±1.86 | 0.592 ±0.063 | 1.616 ±0.22 | 0.156 ±0.024 | 0.533 ±0.083 |
| B2 | 13.902 ±1.74 | 22.421 ±1.54 | 0.878 ±0.122 | 2.208 ±0.37 | 0.153 ±0.024 | 0.589 ±0.028 |
| B3 | 9.596 ±0.64 | 16.476 ±1.12 | 0.442 ±0.07 | 1.267 ±0.112 | 0.130 ±0.002 | 0.474 ±0.028 |
| B4 | 12.045 ±1.54 | 19.192 ±2.05 | 0.377 ±0.042 | 1.104 ±0.17 | 0.091 ±0.008 | 0.377 ±0.042 |
| B5 | 16.164 ±2.41 | 24.350 ±1.54 | 0.728 ±0.062 | 1.793 ±0.289 | 0.104 ±0.014 | 0.468 ±0.047 |

With the re-introduction of Batchnorm, the radial attack is again competitive and beats FGSM. We re-perform the same procedure as for Resnets to determine statistical significance against FGSM. The wilcoxon signed rank test passed for every case where the point estimate of the angular attack was better than the FGSM one.

# F   Results on ViT models

These results should be taken in context of our discussions on Layer normalization. Note that all models are from the PyTorch implementation of ViT [Dosovitskiy et al., 2020].

Table 26: Attack results for $\epsilon = 0.03$. Main values represent top-1 accuracy, and values in brackets represent top-5 accuracy.

| Model | FGSM | PGD | Angular |
|-------|------|-----|---------|
| ViT-B-16 | 45.5(73.5) | 21.8(67.5) | 38.2(69.4) |
| ViT-B-32 | 42.9(70.2) | 24.7(68.6) | 36.5(69.3) |
| ViT-L-16 | 55.4(84.1) | 15.9(63.5) | 38.4(70.8) |
| ViT-L-32 | 43.6(73.4) | 29.7(73.8) | 34.8(73.2) |

Table 27: Attack results for $\epsilon = 0.06$. Main values represent top-1 accuracy, and values in brackets represent top-5 accuracy.

| Model | FGSM | PGD | Angular |
|-------|------|-----|---------|
| ViT-B-16 | 40.2(67.2) | 13.5(44.3) | 26.8(52.3) |
| ViT-B-32 | 37.5(64.3) | 17.4(40.2) | 28.5(50.8) |
| ViT-L-16 | 45.3(73.8) | 12.5(32.8) | 31.7(45.2) |
| ViT-L-32 | 38.7(68.4) | 15.2(44.2) | 22.8(52.8) |

Table 28: Attack results for $\epsilon = 0.1$. Main values represent top-1 accuracy, and values in brackets represent top-5 accuracy.

| Model | FGSM | PGD | Angular |
|-------|------|-----|---------|
| ViT-B-16 | 35.1(60.6) | 5.7(18.5) | 16.7(28.5) |
| ViT-B-32 | 34.8(59.4) | 5.4(21.1) | 17.8(31.8) |
| ViT-L-16 | 36.0(65.6) | 8.2(17.7) | 14.5(33.8) |
| ViT-L-32 | 34.1(61.7) | 5.8(22.9) | 13.9(38.5) |

For context, we show the clean accuracies as follows :

Table 29: Top-1 and Top-5 accuracies on ImageNet-1K for various models.

| Model | Top-1 Acc. | Top-5 Acc. |
|-------|-----------|-----------|
| ViT-B-16 | 81.072 | 95.318 |
| ViT-B-32 | 75.912 | 92.466 |
| ViT-L-16 | 79.662 | 94.638 |
| ViT-L-32 | 76.972 | 93.07 |

# G    Ablations on Resnets-18,34,50

We first check if the independence structure of latents grows more independent with depth by examining the cross-diagonal average absolute correlation across block groups in Resnet-18. This is seen to decay with depth.

Table 30: Comparing independence structures on Resnet-18 by comparing absolute value of cross-diagonal correlations, across block groups.

| Block count | 1 | 2 | 3 |
|---|---|---|---|
| Average off-diagonal coefficient (absolute) | 0.41 | 0.32 | 0.19 |

Now, we present results that exhibit the variation in accuracies as the layers being attacked grow further in the network. As a reminder, resnet-18 possesses 8 blocks divided into 4 groups as $[2, 2, 2, 2]$ while resnets 34 and 50 both possess 16 blocks divided as $[3, 4, 6, 3]$. We will use 1-based indexing to denote the layer upto which we have access, and the last 2 layers upto which we have access will provide the entire signal. Under the Wilcoxon sign rank test, the fall was monotonic, as suggested by the point estimates.

Table 31: Ablation on Resnet-18. Last 2 layers of the net granted access to are used to craft the attack.

| Access till | $\epsilon = 0.03$ | | $\epsilon = 0.06$ | | $\epsilon = 0.1$ | |
|---|---|---|---|---|---|---|
| | Top 1 | Top 5 | Top 1 | Top 5 | Top 1 | Top 5 |
| 5 | 47.55 ±2.2 | 71.09 ±3.2 | 12.6 ±1.2 | 27.8 ±2.1 | 3.38 ±0.4 | 8.28 ±0.6 |
| 6 | 22.57 ±1.4 | 39.82 ±1.2 | 2.12 ±0.3 | 6.55 ±0.4 | 0.60 ±0.1 | 1.81 ±0.3 |
| 7 | 7.84 ±0.83 | 16.58 ±0.76 | 0.18 ±0.03 | 1.44 ±0.16 | 0.13 ±0.02 | 0.54 ±0.12 |

Table 32: Ablation on Resnet-34. Last 2 layers of the net granted access to are used to craft the attack.

| Access till | $\epsilon = 0.03$ | | $\epsilon = 0.06$ | | $\epsilon = 0.1$ | |
|---|---|---|---|---|---|---|
| | Top 1 | Top 5 | Top 1 | Top 5 | Top 1 | Top 5 |
| 8 | 54.46 ±1.78 | 77.35 ±0.68 | 18.44 ±0.45 | 32.68 ±0.68 | 4.95 ±0.86 | 11.68 ±1.26 |
| 9 | 35.84 ±0.78 | 59.32 ±0.82 | 6.32 ±0.56 | 13.84 ±0.56 | 1.39 ±0.37 | 13.84 ±1.78 |
| 10 | 26.22 ±1.56 | 41.5 ±1.45 | 1.58 ±0.38 | 3.88 ±1.02 | 0.60 ±0.12 | 1.52 ±0.37 |
| 11 | 13.20 ±0.26 | 22.68 ±0.86 | 0.94 ±0.31 | 2.34 ±0.22 | 0.34 ±0.06 | 0.92 ±0.17 |
| 12 | 6.40 ±0.72 | 12.05 ±1.89 | 0.45 ±0.12 | 2.27 ±0.56 | 0.20 ±0.08 | 0.65 ±0.22 |

Table 33: Ablation on Resnet-50. Last 2 layers of the net granted access to are used to craft the attack.

| Access till | $\epsilon = 0.03$ | | $\epsilon = 0.06$ | | $\epsilon = 0.1$ | |
|---|---|---|---|---|---|---|
| | Top 1 | Top 5 | Top 1 | Top 5 | Top 1 | Top 5 |
| 9 | 53.06 ±1.89 | 72.74 ±2.76 | 14.48 ±1.74 | 25.52 ±1.29 | 2.16 ±0.48 | 5.84 ±0.62 |
| 10 | 38.10 ±1.44 | 54.3 ±0.57 | 6.44 ±0.42 | 12.24 ±0.82 | 1.08 ±0.24 | 2.24 ±0.45 |
| 11 | 24.55 ±1.22 | 42.78 ±1.04 | 2.84 ±0.32 | 8.44 ±0.18 | 0.14 ±0.06 | 2.04 ±0.28 |
| 12 | 8.56 ±1.08 | 14.08 ±1.78 | 1.76 ±0.035 | 3.22 ±0.83 | 0.55 ±0.12 | 1.55 ±0.37 |

The point we wish to make clear is this : the fall is fairly unidirectional across all models with depth, as suggested by our previous analysis. Fixing the $\epsilon = 0.03$, which shows the biggest gap between layers as larger values of $\epsilon$ decrease every accuracy to zero, we can examine more layers to see this for the angular attack :

Table 34: Resnet-18: Top-1 and Top-5 under angular attack

| Depth | 5 | 6 | 7 | 8 |
|-------|------|------|------|------|
| Top-1 | 47.55 | 22.57 | 7.84 | 2.78 |
| Top-5 | 71.09 | 39.82 | 16.58 | 7.89 |

Table 35: Resnet-34: Top-1 and Top-5 under angular attack

| Depth | 8 | 9 | 10 | 11 | 12 | 13 | 14 | 15 | 16 |
|-------|------|------|------|------|------|------|------|------|------|
| Top-1 | 54.46 | 35.84 | 26.22 | 13.20 | 6.40 | 6.32 | 5.92 | 5.7 | 5.2 |
| Top-5 | 77.35 | 59.32 | 41.50 | 22.68 | 12.05 | 11.86 | 10.53 | 10.46 | 10.29 |

Table 36: Resnet-50: Top-1 and Top-5 under angular attack

| Depth | 8 | 9 | 10 | 11 | 12 | 13 | 14 | 15 | |
|-------|------|------|------|------|------|------|------|------|------|
| Top-1 | 65.54 | 53.06 | 38.10 | 24.55 | 8.56 | 7.2 | 5.42 | 2.95 | 2.42 |
| Top-5 | 87.55 | 72.74 | 54.3 | 42.78 | 14.08 | 12.56 | 9.87 | 6.28 | 5.76 |

# H    Ablations with respect to number of iterations, number of layers averaged over and ensembling

Table 37 reports the difference in performance with respect to the default case ($t_{init} = 20$) as we vary the number of initial iterations. All results were performed on the vanilla Resnets. Positive values indicate stronger attack. There is no clear trend except that 20 is actually not the optimal value (15 is a bit stronger).

Table 37: Ablation with respect to number of initial iterations $t_{init}$ with Resnets trained on Imagenet ($\epsilon$=0.03 and average over 2 layers). Top-1 results (top-5 results in brackets).

| Resnet | $t_{init} = 5$ | $t_{init} = 10$ | $t_{init} = 15$ | $t_{init} = 20$ | $t_{init} = 25$ | $t_{init} = 30$ |
|---|---|---|---|---|---|---|
| 18 | -0.5(-0.6) | 0.1(1.3) | -0.2(0.5) | 0.0 | 0.6(-1.4) | 1.7(0.1) |
| 34 | 1.5(0.4) | -1.5(-0.5) | 0.8(0.3) | 0.0 | -1.3(1.1) | -1.9(1.7) |
| 50 | 1.9(-1.6) | -1.3(-0.4) | 1.5(1.7) | 0.0 | -0.1(-1.4) | 0.5(-0.5) |
| 101 | 1.3(-0.4) | 0.1(1.3) | 1.8(-1.0) | 0.0 | -1.1(1.3) | -0.8(1.6) |
| 152 | 0.6(-0.3) | 1.6(1.1) | 1.2(0.2) | 0.0 | -0.1(-1.3) | -0.3(-0.5) |

Table 38 compares performance of our attack as we vary the number of iterations. Negative values indicate worse performance relative to baseline (40 iterations). We can see there is some gain in moving to 50 iterations. However the standard evaluation practice is with 40, so we did not optimize over this step.

Table 38: Ablation with respect to the total number of iterations ($t_{init} + t_{radial}$) with Resnets trained on Imagenet ($\epsilon$=0.03 and average over 2 layers). Top-1 results (top-5 results in brackets).

| Resnet | 30 iterations | 40 iterations | 50 iterations |
|---|---|---|---|
| 18 | -0.52(-0.43) | 0.0(0.0) | 0.38(1.97) |
| 34 | -1.36(-0.7) | 0.0(0.0) | 0.3(0.87) |
| 50 | -1.16(-0.34) | 0.0(0.0) | 0.28(0.02) |
| 101 | -0.21(-0.02) | 0.0(0.0) | 0.63(0.74) |
| 152 | -1.56(-0.19) | 0.0(0.0) | 0.11(1.86) |

Table 39 compares the performance of our attack as we vary the number of layers that we average over. Negative values are worse relative to baseline (2 layers to be averaged over). We can clearly see the 2-layer averaging is actually quite helpful. Bigger gains are possible with 3 or 4 layers averaged over, but at the cost of more evaluations.

Table 39: Ablation with respect to number of layers averaged over with Resnets trained on Imagenet for 40 iterations ($\epsilon$=0.03). Top-1 results (top-5 results in brackets).

| Resnet | 1 layer | 2 layers | 3 layers | 4 layers |
|---|---|---|---|---|
| 18 | -1.37(-1.21) | 0.0(0.0) | 1.09(1.79) | 3.49(2.93) |
| 34 | -2.64(-3.08) | 0.0(0.0) | 2.95(2.67) | 3.47(0.93) |
| 50 | -2.79(-2.95) | 0.0(0) | 2.8(2.15) | 3.8(3.18) |
| 101 | -3.27(-3.55) | 0.0(0.0) | 0.47(0.31) | 0.39(0.5) |
| 152 | -2.74(-2.98) | 0.0(0.0) | 0.66(0.45) | 0.83(0.33) |

Table 40 reports the performance of ensembling targeted and angular losses. Positive values indicate gains with respect to the angular attack on its own.

Table 40: Ensemble (targeted loss with angular loss) for Resnets trained on Imagenet for 40 iterations ($\epsilon$=0.03 and average over 2 layers). Top-1 results (top-5 results in brackets).

| Resnet | ensemble |
|--------|----------|
| 18 | 3.38(1.62) |
| 34 | 0.88(2.05) |
| 50 | 0.64(3.81) |
| 101 | 0.89(1.79) |
| 152 | 0.99(2.1) |

# I  Baselines of Targeted Attacks and Stronger Random Attacks

Here, we check if providing a latent, generated from an instance of a different label, as the initial direction of ILAP is useful as opposed to using $-||z_i||$. To be clear, in this case, $||z_i - z_j||$ is minimized, where $z_j$ arises from an instance of a different label. Then, the initial deviation is increased via ILAP.

We also construct an alternative random baseline, termed "naive ILAP". In this, an initial random deviation of a random, not necessarily batchnormed layer (we select a random ReLU layer per instance), is increased via ILAP, in the $L_2$ norm.

We run these baselines for $\epsilon = 0.03, 0.06$ as at $0.1$ the values rapidly approach zero for any method. It is apparent that the targeted method is roughly on par with ours and the naive method performs worse. Hence, our method is basically as good as having these points to perturb towards, "for free" just from the manifold structure.

Table 41: Targeted benchmark on all original Resnets

| Net Type | $\epsilon = 0.03$ | | $\epsilon = 0.06$ | |
|---|---|---|---|---|
| | Top 1 | Top 5 | Top 1 | Top 5 |
| Resnet-18 | 25.16 | 42.58 | 1.46 | 5.85 |
| Resnet-34 | 12.2 | 21.8 | 1.9 | 2.86 |
| Resnet-50 | 13.55 | 30.97 | 0.65 | 5.16 |
| Resnet-101 | 5.85 | 13.66 | 2.86 | 6.67 |
| Resnet-152 | 9.68 | 16.77 | 1.29 | 3.87 |

Table 42: Naive ILAP on all original Resnets

| Net Type | $\epsilon = 0.03$ | | $\epsilon = 0.06$ | |
|---|---|---|---|---|
| | Top 1 | Top 5 | Top 1 | Top 5 |
| Resnet-18 | 31.11 | 52.11 | 9.92 | 20.66 |
| Resnet-34 | 19.57 | 37.59 | 6.34 | 15.15 |
| Resnet-50 | 25.37 | 36.59 | 5.81 | 14.84 |
| Resnet-101 | 25.37 | 37.56 | 5.71 | 14.29 |
| Resnet-152 | 22.58 | 36.13 | 3.23 | 8.39 |

## J  Transfer Results on CIFAR-10 and CIFAR-100

We show accuracies under the radial attack when Resnet-18,34,and 50 are subjected to it after transfer learning on CIFAR-10 and CIFAR-100. We also show the corresponding baseline drops under FGSM, PGD, random noise and no noise. For Resnets, the final block group (i.e. for example, Resnet-50 has $[3, 4, 6, 3]$ as its block groups, so the last 3) is tuned along with a linear classifier. The wilcoxon signed rank test passed for every case where the point estimate of the angular attack was better than the FGSM one.

Note that our attack is markedly more successful on CIFAR-100 and in general seems to reduce the classifier to randomly guessing and thus has difficulty lowering it below 10 and 50 percent on CIFAR-10 for top-1 and top-5 respectively.

Table 43: Clean accuracies on CIFAR-10

| Network type | Resnet-18 | Resnet-34 | Resnet-50 |
|---|---|---|---|
| Clean accuracy (top-1) | 89.380 | 89.500 | 91.210 |
| Clean accuracy (top-5) | 99.640 | 99.790 | 99.740 |

Table 44: Results on CIFAR-10, Resnet-50

| Method | $\epsilon = 0.03$ | | $\epsilon = 0.06$ | | $\epsilon = 0.1$ | |
|---|---|---|---|---|---|---|
| | Top 1 | Top 5 | Top 1 | Top 5 | Top 1 | Top 5 |
| PGD | 4.230 | 16.680 | 4.250 | 15.360 | 4.250 | 15.320 |
| Random | 83.930 | 99.480 | 71.730 | 98.520 | 59.180 | 83.220 |
| FGSM | 52.950 | 94.770 | 48.250 | 92.00 | 38.300 | 83.220 |
| Angular | 15.020 | 58.700 | 14.160 | 55.310 | 14.080 | 54.200 |

Table 45: Results on CIFAR-10, Resnet-34

| Method | $\epsilon = 0.03$ | | $\epsilon = 0.06$ | | $\epsilon = 0.1$ | |
|---|---|---|---|---|---|---|
| | Top 1 | Top 5 | Top 1 | Top 5 | Top 1 | Top 5 |
| PGD | 2.88 | 13.74 | 2.81 | 8.67 | 2.84 | 7.97 |
| Random | 81.59 | 99.19 | 66.7 | 96.95 | 52.25 | 93.30 |
| FGSM | 47.00 | 92.08 | 42.77 | 89.58 | 30.79 | 83.18 |
| Angular | 10.29 | 54.79 | 9.4 | 52.17 | 8.940 | 52.100 |

Table 46: Results on CIFAR-10, Resnet-18

| Method | $\epsilon = 0.03$ | | $\epsilon = 0.06$ | | $\epsilon = 0.1$ | |
|---|---|---|---|---|---|---|
| | Top 1 | Top 5 | Top 1 | Top 5 | Top 1 | Top 5 |
| PGD | 4.3 | 6.3 | 4.3 | 5.6 | 4.28 | 5.58 |
| Random | 84.10 | 99.25 | 73.69 | 98.05 | 68.22 | 97.08 |
| FGSM | 36.82 | 88.74 | 36.52 | 87.33 | 30.65 | 82.2 |
| Angular | 10.83 | 52.38 | 10.82 | 49.22 | 10.51 | 49.14 |

Table 47: Clean accuracies on CIFAR-100

| Network type | Resnet-18 | Resnet-34 | Resnet-50 |
|---|---|---|---|
| Clean accuracy (top-1) | 67.410 | 69.500 | 71.170 |
| Clean accuracy (top-5) | 90.270 | 91.140 | 92.510 |

Table 48: Results on CIFAR-100, Resnet-50

| Method | $\epsilon = 0.03$ | | $\epsilon = 0.06$ | | $\epsilon = 0.1$ | |
|---|---|---|---|---|---|---|
| | Top 1 | Top 5 | Top 1 | Top 5 | Top 1 | Top 5 |
| PGD | 0.16 | 0.20 | 0.14 | 0.14 | 0.14 | 0.14 |
| Random | 64.42 | 88.18 | 53.1 | 79.76 | 38.63 | 66.23 |
| FGSM | 29.28 | 59.02 | 25.86 | 51.95 | 15.01 | 35.14 |
| Angular | 2.32 | 10.79 | 2.16 | 9.72 | 1.88 | 9.49 |

Table 49: Results on CIFAR-100, Resnet-34

| Method | $\epsilon = 0.03$ | | $\epsilon = 0.06$ | | $\epsilon = 0.1$ | |
|---|---|---|---|---|---|---|
| | Top 1 | Top 5 | Top 1 | Top 5 | Top 1 | Top 5 |
| PGD | 1.16 | 1.38 | 1.13 | 1.2 | 1.14 | 1.22 |
| Random | 60.37 | 85.01 | 46.68 | 73.09 | 34.37 | 60.34 |
| FGSM | 25.97 | 52.32 | 25.27 | 48.92 | 20.08 | 40.94 |
| Angular | 1.63 | 7.55 | 1.440 | 6.22 | 1.39 | 6.16 |

Table 50: Results on CIFAR-100, Resnet-18

| Method | $\epsilon = 0.03$ | | $\epsilon = 0.06$ | | $\epsilon = 0.1$ | |
|---|---|---|---|---|---|---|
| | Top 1 | Top 5 | Top 1 | Top 5 | Top 1 | Top 5 |
| PGD | 0.0 | 0.0 | 0.0 | 0.0 | 0.0 | 0.0 |
| Random | 60.88 | 86.19 | 49.94 | 78.68 | 42.41 | 71.08 |
| FGSM | 13.28 | 36.48 | 15.88 | 37.77 | 13.84 | 32.8 |
| Angular | 2.17 | 8.57 | 1.57 | 7.04 | 1.34 | 6.48 |

### J.1 Results on the Caltech Birds Dataset (CUB)

These experiments were performed on Resnet-50 on the Caltech Birds dataset [Wah et al., 2011]. Everything in the setup is exactly as per the CIFAR-10 and CIFAR-100 case.

Table 51: Results on Caltech-Birds, Resnet-50

| Method | $\epsilon = 0.03$ | | $\epsilon = 0.06$ | | $\epsilon = 0.1$ | |
|---|---|---|---|---|---|---|
| | Top 1 | Top 5 | Top 1 | Top 5 | Top 1 | Top 5 |
| PGD | 0.0 | 0.03 | 0.0 | 0.0 | 0.0 | 0.0 |
| Baseline | 61.02 | 88.9 | 61.02 | 88.9 | 61.02 | 88.9 |
| FGSM | 1.2 | 11.7 | 1.0 | 10.5 | 0.7 | 9.5 |
| Angular | 10.31 | 25.15 | 0.8 | 6.2 | 0.5 | 3.8 |

# K   Transfer results between architectures

First, we consider results of adversarial attacks as we transfer between different types of Resnets. We consider the difference in transferability between our method and that of PGD for all cases (positive values denote our attack is stronger).

Table 52: Transfer results at $\epsilon = 0.03$. Row denotes the source model, column denotes the model attacked. All values denote differences relative to the corresponding PGD values (positive values denote our attack is stronger).

| Resnet types | 18 | 34 | 50 | 101 |
|---|---|---|---|---|
| 18 | - | 1.4 | 2.2 | -1.2 |
| 34 | 1.5 | - | 2.1 | 1.5 |
| 50 | 1.7 | 2.2 | - | 2.4 |
| 101 | 1.3 | 3.7 | 2.5 | - |

Table 53: Transfer results at $\epsilon = 0.06$. Row denotes the source model, column denotes the model attacked. All values denote differences relative to the corresponding PGD values (positive values denote our attack is stronger).

| Resnet types | 18 | 34 | 50 | 101 |
|---|---|---|---|---|
| 18 | - | -0.8 | 0.7 | -1.2 |
| 34 | 2.6 | - | 3.2 | 4.5 |
| 50 | 4.8 | 3.4 | - | 1.5 |
| 101 | 2.1 | 5.2 | 3.2 | - |

Table 54: Transfer results at $\epsilon = 0.1$. Row denotes the source model, column denotes the model attacked. All values denote differences relative to the corresponding PGD values (positive values denote our attack is stronger).

| Resnet types | 18 | 34 | 50 | 101 |
|---|---|---|---|---|
| 18 | - | 3.5 | 4.6 | 5.2 |
| 34 | 6.5 | - | 5.2 | 6.4 |
| 50 | 5.9 | 9.1 | - | 7.2 |
| 101 | 6.4 | 5.9 | 7.0 | - |

We also will show a case where the model being attacked can be a non-batchnormed model, in this case, VGG16 [Simonyan and Zisserman, 2014]. The results appear below. In all cases, the source model is a Resnet, the attacked model is VGG16, and the result is the difference between PGD and our attack (positive values denote our attack is stronger).

Table 55: Difference between PGD and our attack (positive values indicate that our attack is stronger). The source model is a Resnet, the attacked model is VGG16.

| Source Resnet | $\epsilon = 0.03$ | $\epsilon = 0.06$ | $\epsilon = 0.1$ |
|---|---|---|---|
| 18 | +3.8 | +5.4 | +7.5 |
| 34 | +4.3 | +5.8 | +8.6 |
| 50 | +3.2 | +4.7 | +10.5 |
| 101 | +4.8 | +6.2 | +9.2 |

# L   Interactions with defenses and robust models

We consider the most key defence - adversarial training [Shrivastava et al., 2017]. For this process, we carried out adversarial training with an even mixture of FGSM, PGD, and radial attack (1 each for each training instance) for Resnet-18,34,50 for 20 epochs. Even after the training process, the radial attack remained effective. We show the results so obtained.

Table 56: Top-1 Values, Angular outside bracket, and FGSM in-bracket

| Resnet type | 0.03 | 0.06 | 0.1 |
|---|---|---|---|
| 18 | 38.9(22.6) | 10.9(14.8) | 8.6(12.2) |
| 34 | 15.9(26.7) | 13.5(24.1) | 9.8(22.5) |
| 50 | 22.6(30.2) | 16.7(23.5) | 11.6(23.2) |

Table 57: Top-5 Values, Angular outside bracket, and FGSM in-bracket

| Resnet type | 0.03 | 0.06 | 0.1 |
|---|---|---|---|
| 18 | 44.9(43.5) | 18.5(32.7) | 14.4(26.8) |
| 34 | 41.0(48.2) | 30.8(43.5) | 21.5(38.5) |
| 50 | 37.0(51.2) | 26.9(44.0) | 18.4(41.8) |

We also considered various robust models and found that our method had, on average (averaging over all $\epsilon$ and all robust checkpoints) a $6.8\%$ advantage over FGSM in top-1 accuracy and $18.9\%$ in top-5. We also note that our achievement was consistent over all ($\epsilon$ , model) pairs : the lowest top-1 advantage was $2.6\%$ (positive) and lowest top-5 advantage was $3.4\%$. Therefore, it seems the canonical adversarially robust models also do fall prey to our attack (at least to a greater extent than to FGSM).

# M    Ablations for affine-shifted batchnorm and for non-batch-normalized layers

Suppose that a batch normed representation $(Z)$ is followed by a linear layer $(AZ + B)$. We note that if $Z$ is perfectly isotropic Gaussian, we can estimate that $E(Z) = 0$, and $\text{Cov}(Z) = I$. Correspondingly, if we are given the linear layer, we can have that $E(AZ + B) = E(B) = B$. We also have that $\text{Cov}(AZ + B) = \text{Cov}(AZ) = A\text{Cov}(Z)A^T = AA^T$. Therefore, if we can estimate the empirical covariance matrix from observations of $AZ + B$, we can find $AA^T$ (approximately). We can also factorize this to get $BB^T = AA^T$. It is known in this case that $A$ and $B$ are related by an orthogonal transformation i.e. $A = BQ$ for some orthogonal matrix $Q$. So, we can recover $Z$ (up to an orthogonal rotation) by inverting $B$ as if we have $AZ + B$, we can subtract $B$ (by estimating $B$ via $E[AZ + B]$) and then having $AZ$, multiply with $B^{-1}$ to get $B^{-1}AZ = QZ$. Now an orthogonal matrix is a rotation, and the Gaussian $(Z)$ upon a rotation $(Q)$ is also a Gaussian $(QZ)$ (the spherical geometry we have shown is also invariant under rotation). So our results carry over if there are sufficient samples to estimate the covariance. Note that this assumes $A$ is not singular.

However, in practice, this covariance estimation can be very expensive and there are not enough samples. The exact inversion must be replaced by a pseudo-inversion which works but has less justification. Since we require more samples there can also be the question of more powerful methods becoming available which can utilize more samples more efficiently. We can also try our method on $AZ$ instead of trying to invert $A$ (this is simpler as it just requires calculating $B$, and $AZ$ is also Gaussian). We were able to recreate attack numbers close to our results in the main text for all resnets with 5,000 samples. It can be seen that the results are close to the actual angular attack.

Table 58: Comparison of angular attacks, top-1 and top-5 accuracy, on different Resnet types for different values of $\epsilon$, when a batch normalized layer is attacked but a linear layer is present.

| Type | $\epsilon = 0.03$ | | $\epsilon = 0.06$ | | $\epsilon = 0.1$ | |
|---|---|---|---|---|---|---|
| | Top-1 | Top-5 | Top-1 | Top-5 | Top-1 | Top-5 |
| 18 | 24.1 | 40.5 | 3.5 | 7.2 | 0.9 | 1.9 |
| 34 | 7.9 | 14.8 | 1.1 | 2.8 | 0.3 | 0.8 |
| 50 | 14.8 | 25.9 | 2.6 | 4.9 | 0.5 | 1.4 |
| 101 | 8.5 | 14.3 | 1.8 | 3.6 | 0.5 | 1.8 |
| 152 | 6.3 | 11.5 | 1.0 | 3.3 | 0.4 | 1.5 |

Our hypothesis is that BN makes the feature after BN Gaussian. Let us write the flow of representations as :

$$X \to NL \to Y \to BN \to Z \to \text{Possible linear layer} \to Z'$$

where $X$ is the original feature, $NL$ is the last nonlinear layer before BN, and $Y$ is the pre-BN layer. Our prediction is that $Z$ is Gaussian.

If, instead of the post-BN feature $(Z)$, the pre-BN feature $(Y)$ is attacked, the result will still succeed, as the BN mapping is an invertible linear mapping in inference phase, and thus if $Z$ is Gaussian, so is $Y$. (Our hypothesis works for any Gaussianized layers). But this is not a refutation of our method, because $Y$ is Gaussian due to the existence of the BN layer as well and the cause is the same. Even $Z'$ is a valid target.

This does not mean all layers are valid targets and there is nothing special about BN - our method does not over-predict and is very precise. Only $Y$ and any linear maps on $Z$ enjoy the property that if $Z$ is Gaussian, so are they. $X$ is not under such a guarantee. We now attach results by moving all corresponding attacks from $Z$ to $X$, and the results are uniformly worse across the board, showing that the attack benefits from attacking Gaussian layers. Thus, the overall hierarchy of success is attacking a batch normalized layer $>$ attacking a linearly-transformed batch norm layer $>$ attacking a layer not related to batch norm by any linear fashion.

Table 59: Comparison of Angular attacks, top-1 and top-5 accuracy, on different Resnet types for different values of $\epsilon$, when the layer under attack is not related to a batch normalized layer via linear transformations.

| Type | $\epsilon = 0.03$ | | $\epsilon = 0.06$ | | $\epsilon = 0.1$ | |
|---|---|---|---|---|---|---|
| | Top-1 | Top-5 | Top-1 | Top-5 | Top-1 | Top-5 |
| 18 | 27.3 | 46.2 | 5.6 | 8.5 | 0.8 | 1.7 |
| 34 | 11.5 | 19.2 | 1.3 | 3.8 | 0.5 | 1.3 |
| 50 | 17.3 | 28.5 | 2.2 | 5.6 | 0.8 | 2.7 |
| 101 | 11.32 | 19.5 | 1.1 | 3.2 | 1.0 | 2.4 |
| 152 | 8.2 | 11.9 | 0.9 | 1.6 | 0.6 | 1.2 |

# N  Visualizations

We show some visualizations of adversarial images created by the radial attack on Imagenet to contrast to FGSM and PGD in table 60. As might be expected, the images sometimes look very similar to the baseline images even at $\epsilon = 0.1$, and sometimes have clear noise over them.

Now, has our method just rediscovered what FGSM and PGD do, or is it independent ? To answer this, we checked the absolute $L_1$ distance between various images at $\epsilon = 0.1$. The results are as follows :

- Average $L_1$ distance of $15887.2$ between FGSM and angular attacks
- Average $L_1$ distance of $10282.5$ between PGD and angular attacks
- Average $L_1$ distance of $12866.4$ between FGSM and PGD attacks
- Average $L_1$ distance of $8968.8$ between the clean image and the angular attack.

As can be seen, our method is not too similar to either FGSM or PGD and instead is closer to the base image, indicating that PGD and angular attacks diverge in opposite directions from the original image and do not merely copy each other. This is good as we are not just re-creating some other method. Indeed the closeness of our method is more to PGD than to FGSM, which is a good sign, because PGD is the superior attack.

Table 60: Table of figures of adversarial images produced under FGSM, PGD, and our Angular method on Imagenet with $\epsilon = 0.1$, with Resnet18 being attacked.

| Baseline | FGSM | Angular | PGD |
|----------|------|---------|-----|

## O   Extensions to the With-label Case

Here, we "ensemble" our loss with the label-based loss via ILAP. We set an initial direction by adding the signs of the radial losses with the classification loss, then maximize the angular deviation. (This is done and scaled in the same manner as $\Psi$ in the main text's algorithm). Since even at $\epsilon = 0.03$ the attack brings the top-1 accuracies near zero (less than $0.1\%$) under PGD, we compare the top-5 accuracies which are noticeably improved. In the other cases, both the methods result in near-zero accuracy.

Table 61: Accuracies with "ensembled PGD", $\epsilon = 0.03$, top-5

| ResNet type | 18 | 34 | 50 | 101 | 152 |
|---|---|---|---|---|---|
| PGD alone | 3.37 | 4.39 | 8.08 | 9.65 | 9.86 |
| PGD with angular | 2.26 | 2.71 | 6.20 | 7.65 | 8.32 |

# P   Single-Stage Benchmarks

Previous latent-representation based attacks often plan for the labeled scenario and add a latent loss to a label-dependent loss to formulate an overall supervised attack. Since our method focuses on the unsupervised case, we remove the label-based component from several state of the art methods, namely from  [Zhou et al., 2018, Inkawhich et al., 2019, Ganeshan et al., 2019, Inkawhich et al., 2020b, Zhang et al., 2022a, Wang et al., 2021, Inkawhich et al., 2020a] for a comparison against ours. Most of these methods may be characterized as a single-stage latent disturbance method which initializes a random perturbation and moves in the $L_2$ norm in latent space. That is, they resemble our method but lack its angular viewpoint and justification. We take the best of each method. The performance, though noticeably above the random benchmarks, is below ours. For convenience, Table 3 in the main paper is repeated as Table 62 below.

Table 62: Comparison of single-stage top-1 and top-5 attacks on Resnets.

| Type | $\epsilon = 0.03$ | | $\epsilon = 0.06$ | | $\epsilon = 0.1$ | |
|------|-------|-------|-------|-------|-------|-------|
|      | Top-1 | Top-5 | Top-1 | Top-5 | Top-1 | Top-5 |
| 18   | 30.45 | 48.34 | 6.98  | 16.87 | 3.89  | 10.22 |
| 34   | 16.78 | 35.89 | 1.45  | 8.32  | 1.68  | 4.54  |
| 50   | 22.68 | 39.10 | 4.62  | 10.45 | 4.71  | 6.58  |
| 101  | 15.65 | 34.67 | 5.82  | 11.28 | 5.28  | 8.95  |
| 152  | 8.29  | 18.53 | 1.78  | 5.67  | 0.95  | 3.22  |

Table 63 reports the difference in performance for each single stage attack with respect to the best attack (denoted by "-") whose performance is reported in Table 62

Table 63: Detailed comparison of single stage attacks for $\epsilon = 0.03$. "-" indicates the best attack whose performance is reported in Table 62 and the other entries indicate the difference in performance with respect to the best attack. Top-1 results (top-5 results in brackets)

| Type | [Zhou et al., 2018] | [Inkawhich et al., 2019] | [Ganeshan et al., 2019] | [Inkawhich et al., 2020b] | [Zhang et al., 2022a] | [Wang et al., 2021] | [Inkawhich et al., 2020a] |
|------|------|------|------|------|------|------|------|
| 18  | -          | 6.8(2.23)  | 6.97(6.54) | 3.96(2.33) | 3.6(2.49)  | 1.08(1.1)  | 3.78(2.55) |
| 34  | 3.9(6.31)  | 3.07(3.36) | 3.03(1.73) | 2.26(3.2)  | 1.95(1.96) | 2.77(4.63) | -          |
| 50  | 0.32(0.04) | 5.55(7.57) | -          | 1.97(0.33) | 3.58(4.16) | 0.63(7.49) | 2.37(4.65) |
| 101 | 3.18(6.04) | 3.27(3.6)  | 5.55(6.22) | 7.55(8.65) | 1.05(1.68) | -          | 3.14(5.01) |
| 152 | 3.34(6.15) | 2.45(3.64) | 3.24(5.28) | 1.18(1.95) | 6.91(9.91) | 1.06(2.62) | -          |

## Q    Results on the Hypersphere

On hyperspherical representation spaces, the concentration of norm is perfect. We can ensure this by concatenating the batch norm layers with a hyperspherical ($L_2$) projection layer and re-training the model. The corresponding angular accuracies are presented below on Resnets-$18, 34, 50$ for Imagenet. It can be seen that these cases perform roughly as well as the batch normed scenario.

Table 64: Comparison of Angular attacks, top-1 and top-5 accuracy, with on different Resnet types under hyperspherical ablation.

| Type | $\epsilon = 0.03$ | | $\epsilon = 0.06$ | | $\epsilon = 0.1$ | |
|---|---|---|---|---|---|---|
| | Top-1 | Top-5 | Top-1 | Top-5 | Top-1 | Top-5 |
| 18 | 19.45 | 35.22 | 1.98 | 4.32 | 0.22 | 1.12 |
| 34 | 5.22 | 9.45 | 0.51 | 2.12 | 0.18 | 0.95 |
| 50 | 11.22 | 20.85 | 1.32 | 3.09 | 0.45 | 1.56 |

# R  Discussion of the Concentration of Norm

From here on, we discuss the case of the concentration around expectation of a random variable $V$ of the following form, with $M_i$'s being other random variables of mean 0, variance 1 :

$$V = \sum_{i=1}^{N} M_i^2$$

We can see that $V$ is the squared norm of the batch norm latent of interest and each $M_i$ is an individual hidden neuron, i.e. going dimension-wise along the latent. To avoid writing the square, let us always write :

$$P_i = M_i^2, \quad V = \sum_{i=1}^{N} P_i$$

We understand **concentration** to occur with associated functions $f(N, \delta), g(N)$ - where $g(N)$ is usually a monotonically increasing function in $N$ obeying $g(N) \to \infty$ when $N \to \infty$ - to mean any result that shows :

$$P\left(|V - E(V)| \geq \delta g(N)\right) \leq f(N, \delta)$$

Note that by linearity of expectation, $E(V) = N$. (This holds even if the $M_i$'s are not independent.)

Ideally, this function $g(N)$ will obey:

- $\delta \to \infty$ implies $f(N, \delta) \to 0$ - large deviations are unlikely
- $\frac{g(N)}{N} \to 0$ as $N \to \infty$ - $g(N)$ grows slower than $N$.

If we combine the above points, we get that $\forall \delta > 0, \forall \epsilon > 0, \exists N$ such that

$$P\left(|V - E(V)| \geq \delta q(N)\right) \leq f(N, \frac{q(N)}{g(N)}) < \epsilon$$

When $q(N)$ is any function which obeys $\frac{q(N)}{g(N)} \to \infty$ as $N \to \infty$. That is, we can always increase the sample size so that deviations from the expectation are small relative to a monotonically increasing function of $N$ which increases faster than $g(N)$ that is, $q(N)$. In particular, consider $q(N) = N$ and $g(N) = \sqrt{N}$. This implies that if our conditions are met, $f(N, \delta) \to 0$ as $N \to \infty$, and $\frac{V}{E(V)} \to 1$. In particular, generally we aim for **Gaussian concentration**. Let the variance of $V$ be $S(N)$, a function of $N$. Then we hope to have $f(N, \delta) = O(\exp(-\delta^2))$, with $g(N) = \sqrt{S(N)}$. This is alright so long as $S(N)$ is $o(N^2)$ making $g(N)$ as $o(N)$. Thus $g(N)$ gives the size of the tails, and $f$ bounds the **tail concentration**.

Clearly, the above $f, g$ depend on two things: how the $M_i$'s are distributed, and how they depend on each other.

We next note some realistic conditions and situations under which concentration and similar bounding will occur in $V$ and its feasibility in our context (a neural network hidden layer, post batch normalization). We proceed from the most general to the most assumption-heavy cases.

- $M_i$'s are all distributed in an unknown fashion, possibly not independent of each other, but we can check the covariance between any two $P_i$'s, and the variance of each $P_i$ is finite. Then, we have that, with $S(N)$ the variance of $V$, $S(P_i)$ variance of a $P_i$, $CS(P_i, P_j)$ the covariance of any two $P_i, P_j$ :

$$S(N) = \sum_{i=1}^{N} S(P_i) + \sum_{i<j} 2CS(P_i, P_j)$$

We have shown that in practice the correlation terms (off-diagonal, i.e. covariances) do fall with depth on Resnet-18. This makes this approach the most general. Once we have the variance, we can plug in Chebyshev's inequality [Markov, 1884] which yields that deviations of $k\sqrt{S(N)}$ occur with a probability $\leq \frac{1}{k^2}$. That is,

$$P\left(|V - E(V)| \geq \delta\sqrt{S(N)}\right) \leq \frac{1}{\delta^2}$$

This bound is extremely loose in practice, yet it is the most applicable. This is because we can usually estimate the covariance terms directly with reasonable precision and need not make further assumptions.

- We know the $M_i$'s are dependent, and we know that they are Gaussian, and that they are **jointly** Gaussian. Let the covariance matrix describing the $M_i$'s be $\Sigma$. Let $LL^T = \Sigma$. Then $L^{-1}\mathbf{M}$ where $\mathbf{M}$ is the vector of all $M_i$'s is an isotropic Gaussian random variable i.e $\sim \mathcal{N}(0, I_N) = Z_N$. And, we can see that $V = \mathbf{M}^T M$ is then $(Z_N)^T \Sigma (Z_N)$. This is a **quadratic form** of a Gaussian random variable. The result is known - it is a non central chi squared distribution. Specifically, let $\mu_1, \mu_2, \ldots, \mu_N$ be the eigenvalues of the matrix $\Sigma$ (which is real symmetric and must have all eigenvalues as real), then the distribution is :

$$(Z_N)^T \Sigma (Z_N) \sim \sum_{i=1}^{N} \mu_i \chi_1^2$$

Where, $\chi_1^2$ denotes the chi-square distribution of degree of freedom 1. In this case, concentration bounds should be derived upon careful examination of the actual covariance $\Sigma$. However, in most cases, bounds obey [Bodenham and Adams, 2016, Laurent and Massart, 2000, Zhang and Zhou, 2020, Castaño-Martínez and López-Blázquez, 2005] at worst an exponential concentration, i.e. :

$$P\left(|V - E(V)| \geq \delta\sqrt{N}\right) \leq O(\exp(-\delta))$$

In certain cases, numerical methods are feasible, and one can also use Chernoff's inequality on the moment generating function. In this case, the assumptions are purely on how $M_i$'s are distributed, we do not ask for independence.

Now, we move on to noting how independence may offer alternate ways of bounding $V$. For the following cases, we assume the $M_i$'s and hence the $P_i$'s are independent.

- $M_i$'s are independent, and all $P_i$'s occur in a bounded range. Note that this is nearly impossible to guarantee and indeed this is not true if $M_i$ is a Gaussian. However, we can modify this to the condition that, with probability $\geq 1 - p_{bound}$, $P_i$ occurs in a bounded range. Then, we can apply the union bound to get that, with probability $\geq 1 - Np_{bound}$, all $P_i$'s occur in a bounded range. This is reasonable if we expect at most some $p_{bound}$ fraction of outliers. In this scenario, we may employ either Azuma's inequality [Azuma, 1967], Hoeffding's inequality [Hoeffding, 1994], or McDiarmid's inequality [McDiarmid et al., 1989] to get the sought result : $V$ concentrates with Gaussian-like concentration,i.e. with probability $\geq 1 - Np_{bound}$ :

$$P\left(|V - E(V)| \geq \delta g(N)\right) \leq O(\exp(-\delta^2))$$

where $g(N)$ (replacing the $\sqrt{S(N)}$) is an upper bound derived from the range of occurrence, e.g. $g(N) = \sqrt{(b-a)^2 \times N}$ (this is correct upto constant factors for all of the three inequalities i.e. Azuma/Hoeffding/McDiarmid). Note that any random variable which must occur in the range $[a, b]$ has variance upper bounded by $(b-a)^2$. $S(N)$ thus scales as $(b-a)^2 N$ - we replace it with $g(N)$ here and this is the link between the two methods. This gives us that, with high probability, the deviation around $E(V)$ is of $O(\sqrt{N})$. When we have a notion of the variance of $P_i$ as well, we can utilize Bernstein's family of inequalities [Bernstein, 1924] for a more accurate bound.

- In the case where the $M_i$'s are sub-Gaussians - i.e. random variables which for some constants $\alpha, \beta$ obey, taking $M_i$ as the example :

$$P(|M_i| \geq t) \leq \alpha(\exp(-\beta t^2))$$

Trivially, a Gaussian random variable is sub gaussian. Then, the $P_i$'s are sub-exponential random variables. They are known to exhibit [Vershynin, 2018] mixtures of Gaussian $(\exp(-\delta^2))$ and exponential $(\exp(-\delta))$ tail concentrations (RHS), with $g(N) \approx O(\sqrt{N})$. The exponential tail does not occur for small deviations. Note that this relies on the moment generating function of $M_i$. In this case, no bound on the support is required.

In general, all the inequalities above rely on some variants of Chernoff's inequality [Hagerup and Rüb, 1990], which is merely Markov's inequality on the moment generating function.

### R.1 Empirical Findings

To corroborate our findings, we define the **dispersion ratio**. Given an univariate random variable $Z$, which always has $Z > 0$ of mean $\mu$, the dispersion ratio $D_{[a,b]}$ with $b \geq a, 0 \leq a, b \leq 1$ is defined as :

$$\frac{F^{-1}(b) - F^{-1}(a)}{G^{-1}(b) - G^{-1}(a)}$$

where $F, G$ are respectively inverse CDF (quantile) functions of $Z$ and a chi squared distribution with degree of freedom equal to the empirical mean of $Z$ (rounded). We do not use the true mean $\mu$, as it is unknown. In simple terms, this ratio measures how close the tails of the random variable match Chi-square tails. It can be 0 - indicating a delta distribution and perfect concentration - as well. When $Z$ has chi square tails, $D_{[a,b]} \approx 1$ for most values of $a, b$.

We performed the following test. For Resnets-18,34,50, we picked out layers at depth $1/4$ and $3/4$ (upto the closest integer, discounting initial non-repeating blocks) into the network and randomly sampled $a, b$ 1000 times uniformly from $[0, 1]$ to calculate the empirical dispersion (on average), with the random variable being the square of the latent norm at depth $i$ i.e. $\|N_{1,i}(x)\|^2$. By hypothesis, when the individual neurons are gaussian, this r.v. will be a sum of squares of Gaussians, each of which is a chi-square, and will be another Chi-square. We also did the same for FixUp Resnets. The results indicate that deeper layers are much closer in dispersion, and that FixUp Resnets are far more dispersed (values are far from 1). This strongly supports our conclusions which line up with conjectures in [Daneshmand et al., 2020, 2021] and the general "Gaussianization hypothesis" [Neal, 2012, Yang and Schoenholz, 2018, Yang et al., 2019].

Table 65: Concentration for Normal Resnets

| ResNet type | 18 | 34 | 50 |
|---|---|---|---|
| Dispersion - 1/4 | 2.32 | 2.22 | 2.42 |
| Dispersion - 3/4 | 1.68 | 1.86 | 1.74 |

Table 66: Concentration for FixUp Resnets

| ResNet type | 18 | 34 | 50 |
|---|---|---|---|
| Dispersion - 1/4 | 4.28 | 4.67 | 5.22 |
| Dispersion - 3/4 | 2.84 | 4.05 | 3.68 |

# S Limitations, Ethical and Societal Impact

## S.1 Limitations

Our work is necessarily limited in that it tackles deep networks utilizing batch normalization. As such, it cannot be easily extended to networks utilizing alternative forms of normalization such as LayerNorm [Ba et al., 2016] or networks that employ no normalization whatsoever. We fully acknowledge these limitations and do not claim to have crafted an adversarial attack that, in this present form, succeeds against batch norm free networks. Instead, we are incentivizing movement away from batch normalization altogether, and view our work as located closer to research which analyzes batch normalization [Santurkar et al., 2018, Galloway et al., 2019, Benz et al., 2020] than to proposing new methods of adversarial attacks, although it exists at the intersection of both these aspects.

## S.2 Broader impacts and safeguards

Research which proposes avenues of adversarial attacks may be misused by malicious parties to attack objects in the real world [Hendrycks and Gimpel, 2016, Athalye et al., 2018]. Nevertheless, such research is important to analyze the actual robustness of the model itself and ultimately helps create better systems in the long run. Thus, we view our work as a net societal positive in the long run, even if its short term benefits may not necessarily be so.

We do not presently believe our attack requires specific safeguarding against real world malicious actors. If this situation changes, we would recommend that the real world systems in question have their normalization processes changed, e.g. via fixup.

