# OpenReview forum: "Batchnorm Allows Unsupervised Radial Attacks"
_NeurIPS.cc/2023/Conference — NeurIPS 2023 poster_

### Official Review · Reviewer_ykjk · 2023-07-01

**Soundness:** 3 good
**Presentation:** 2 fair
**Contribution:** 2 fair
**Rating:** 4
**Confidence:** 3

**Summary:**

This paper shows that for batch normalized deep image recognition architectures, intermediate latents that are produced after a batch normalization step by themselves suffice to produce adversarial examples using an intermediate loss solely utilizing angular deviations, without relying on any label. The success of the proposed method implies that leakage of intermediate representations may create a security breach for deployed models, which persists even when the model is transferred to downstream usage. The proposed attack also empirically works even for transformer architectures (ViT) that employ Layernorm over batchnorm.

**Strengths:**

1.	The proposed method is well motivated and technically sound.

The proposed loss is motivated by the geometry of batch normed representations and their concentration of norm on a hypersphere and distributional proximity to Gaussians. Through theoretical analysis, the authors show the contribution of batch normalization to an unsupervised radial attack and the optimal layers for the latent to lie near the end of the network. These theoretical findings are also supported by their experimental results. Therefore, the proposed method is well motivated and technically sound.

2.	The experimental results are promising.

Comparing to single-stage attacks, the proposed method can obtain higher attack success rates, which shows the effectiveness of the proposed attack method and loss function.


**Weaknesses:**

1.	The significance of the proposed method is not clear.

While the proposed method can get rid of the reliance on label information, as shown in their experiments, the obtained attack success rates are inferior to well-recognized attacks, i.e., PGD. Therefore, it is unclear what is the significance of the proposed method. Since the authors consider white-box settings, attackers can use white-box attacks, like PGD, with predicted labels of the target images, which can also avoid the usage of ground-truth label information. A potential application of the proposed method seems to be the transfer learning setting. However, from the experimental results, i.e., Table 10, the proposed method is also inferior to PGD. Therefore, it is unclear what the proposed method can be used for.



2.	Some claims are not validated or clearly explained.

1)	The authors failed to clearly explain the meaning of Figure 1, which is closely related to their motivation. The notation of Figure 1 is different from that in the text, which can be confusing.
2)	The authors claimed that “the 2-step process is key.” However, they failed to explain why the proposed 2-step process is better than 1-step methods.
3)	The meaning of lines 171-173 is confusing, can the authors further clarify their claims?
4)	When we have a different latent Zj from a point Xj of a different label from Xi, the authors proposed to use this latent in their loss function to generate adversarial samples. However, this scheme is inferior to the proposed method. Can the authors explain why?

3.	Presentations can be improved.

1)	The authors split the attack results of their method and baseline methods into different tables, i.e., Tables 1-4 and Table 8, which makes it inconvenient to compare different attacks. The authors can merge these tables for better readability.
2)	The authors claimed one of their contributions as “To improve the supervised attack case when labels are available by using our loss in conjunction with the loss from the true label.” However, the corresponding experimental results are put into the appendix. This reviewer would suggest the authors put these experimental results into the main paper to support their claim.


**Questions:**


1.	What can the proposed method be used for?

2.	Why is the proposed 2-step process better than 1-step methods?

3.	The meaning of lines 171-173 is confusing, can the authors further clarify their claims?

4.	When we have a different latent Zj from a point Xj of a different label from Xi, the authors proposed to use this latent in their loss function to generate adversarial samples. However, this scheme is inferior to the proposed method. Can the authors explain why?


**Limitations:**


The authors have acknowledged the limitations of their work that it cannot be easily extended to networks utilizing alternative forms of normalization such as LayerNorm or networks that employ no normalization whatsoever. It is acceptable that this study focuses on a specific but widely used module in modern neural networks. Please see the weaknesses section for the other limitations of this work.

The authors have discussed the potential negative societal impact of their work. The authors also proposed a mitigation strategy to alleviate potential harm of this study.

---

> ### Author Rebuttal · Authors · 2023-08-09
>
> Thank you for the review, our comments in response are as follows to the weaknesses :
>
> Weakness 1: The proposed method is most suitable for the cases where a feature extractor (known to us) has been used to train a model (unknown to us). Generally in these cases the first K layers of the new fine-tuned model are left unchanged. Hence, we must attack this model without knowing anything except what the first K layers are. In fact, in these cases, not even a predicted label is usable, as the feature extractor does not emit labels but representation vectors. This case arises naturally in scenarios like :
>
> - Knowing a model is fine-tuned from e.g. Resnet => the first few layers are likely to be unchanged, thus this applies
> - Knowing a model uses a ready-made representation model and then trains on top of it
>
> Our attack is successful in that scenario, whenever these first K layers utilize batchnorm or Layer norm, which is a significant fraction of all models. We believe this scenario to be more and more common as pre-trained and fine-tuned models become the norm instead of training from scratch. Note that in this case, PGD is not applicable as we do not know the full new model and cannot even get soft labels. In fact, we may not even know the new set of labels (when we pre-train on Imagenet, and we are given a new model that is for CIFAR-10, we can use our attack without knowing that it is CIFAR-10, or it has 10 categories, but the soft label approach requires this knowledge of what the label structure of the downstream dataset is)
>
> 2.1) We apologize for the oversight. The notation is different as the complete notation had not been introduced at this point in the paper, however, we believe that the figure should be as early as possible for introducing visual intuition to the reader. We will move the notation earlier and align the caption going forward.
>
> 2.2) For our attack, we explicitly state why the simple 1-stage solution does not work under angular similarity - If the angular loss is used directly, the exact hyper-spherical condition has a differentiability problem (Section 2.1). Of course, *other* attacks might be 1-stage, but they are empirically worse, as we show in our comparison to single-stage attacks (appendix O)
>
> 2.3) Regarding lines 171-173 : Suppose we are in a simple regime where we have an input $X$ with output either 0 or 1. For simplicity, say $X_1$ is mapping to 0. We are told that all valid inputs $X$ lie on a hypersphere, and that opposing points on the hypersphere are the most dissimilar, and that given an $X_1$, we must find $X_2$ with a different label i.e. 1. In the batch normed regime, the center of the hypersphere is the origin. In the non batch normed regime, there is not enough information to solve the problem, but in batch normed regime, we can reflect $X_1$ to get $X_2 = -X_1$ as our candidate for label 1. The point of the lines is to show batch norm gives us one more point of information which can be enough information leakage to make the robustness weaker for toy cases.
>
> 2.4) Consider the toy case of 2.3, and suppose the labels are one of : 0, 0.01, and 1 (Assume that the similarity of the labels reflects the numerical gap). In this case, given a point of label 0, the random point of a different label might be one with label 0.01. Due to high similarity of the two labels, there is no effective movement. When points of different label are being drawn at random, there is no way to check against this. Thus, under the case where $-X_1 = X_2$ is the most dissimilar point to $X_1$ and all similarity metrics are angular, a differently labeled $X_3$ to $X_1$ might result in only a small angular movement vs. moving towards $X_2 = - X_1$. Especially when $X_3$ and $X_1$ are of closely related classes (e.g. cat and dog vs airplane), the projection $- X_1 = X_2$ is a more effective target to move to vs randomly selecting an element of another class.
>
> 3.1) We will certainly merge these tables. Thank you for the catch.
>
> 3.2) These results will be put in the main paper instead.
>
> We hope this addresses the questions / weaknesses. Please let us know if these address your concerns. Note that the questions have been answered under weaknesses (Question 1 as weakness 1, Q2 as 2.2, Q3 as 2.3, Q4 as 2.4).
>
> With respect to your "Limitations" section, we would also like to note that Layernorm is empirically vulnerable to our attack and analyzed in the appendix for our method.
>
> We hope we have addressed your concerns in full and resolved your issues with the paper - if not, please let us know.

---

> > ### Author Response · Authors · 2023-08-17
> > **Follow-up response**
> >
> > Dear reviewer, please let us know if the above rebuttal satisfied your queries, since the discussion period will now draw to a close. We would like to respond to any further queries or clarifications before then.

---

> > ### Comment · Reviewer_ykjk · 2023-08-21
> >
> > Thanks for your detailed response. Most of my questions have been clarified.
> > However, I think that the claimed use case of the proposed method is still unrealistic. The attacker needs to know the exact feature extractor that the victim uses. During fine-tuning, the victim also needs to keep the feature extractor unchanged. More importantly, in the transfer learning setting (Table 10), the proposed method is also **inferior** to PGD.  Besides, I think the authors should add all these clarifications and experiments requested by the reviewers to their paper to improve its contribution and presentation. Therefore, I will keep my score.

---

> > > ### Author Response · Authors · 2023-08-21
> > > **Reply to the response**
> > >
> > > Dear reviewer - knowing the exact feature extractor (up to the initial layers) is not at all an unrealistic scenario and is much easier to guess/leak than the actual weight matrices of the feature extractor, which is infinitely more complicated. A large fraction of vision models use Resnet, ViT etc. as a feature extractor in the sense of a fine-tuned base, and in NLP, the same had happened with BERT embeddings and now GPT models. (In fact, for recent large language models, one can even now *directly ask* the model if it is a language model trained by OpenAI, etc. to leak what kind of model it was initially fine-tuned from). Note that our adversarial examples also transfer well (appendix J) and it will suffice to simply know the family (i.e. Resnet) over the specific model (i.e. Resnet-18) which is quite a plausible leakage model.
> > >
> > > Moreover, during fine-tuning, we are *not* saying that the entire feature extractor is unchanged, but that the first $k$ layers of the model used remain unchanged and frozen, which is actually in line with what happens, as generally only the last layers are fine-tuned in the pre-train setting.
> > >
> > > Finally, the PGD advantage arises in transfer learning for the case where the full model is known. It is not in fact directly comparable to our method at all as it uses the full model, while we do not. The PGD attack is not put in as a fair competitor to the partial knowledge scenario discussed above and has only been put in to demonstrate a ``ceiling". So, in fact, there is no inferiority to PGD - the two attacks occupy different roles and work with different required information.
> > >
> > > We will put all this in the paper in terms of content. We hope this further clarifies your doubts.

---

> ### Comment · Area_Chair_AyQx · 2023-08-20
> **Reminder from AC**
>
> Dear reviewer,
>
> The author-reviewer discussion period ends in 2 days. Please review the authors' rebuttal and engage with them if you have additional questions or feedback. Your input during the discussion period is valued and helps improve the paper.
>
> Thanks, Area Chair

---

### Official Review · Reviewer_7ce2 · 2023-07-05

**Soundness:** 3 good
**Presentation:** 2 fair
**Contribution:** 3 good
**Rating:** 6
**Confidence:** 3

**Summary:**

In this paper, the authors proposed a label-free attack utilizing a portion of all layers, which does not require to have gradient access to the full model, and the generated adversarial methods generalize to the case where the model was fine-tuned afterwards. These results have relevance at the intersection of the theory and practice of adversarial robustness and the growing study of batch normalization and its weaknesses and drawbacks.

**Strengths:**

The proposed attach method is a label-free method, and it is a quite strong one. Moreover, the authors use the diagram to demonstrate the idea on their proposed method, which is very clear. The paper is well-written, and easy to follow. Moreover, the authors extended their work to the layer normalization case, showing the proposed method is a quite generic one.



**Weaknesses:**

The proposed method only tested in the white-box setting, which means the adversary has the access to all parameters of the model, while in practice, this might be not practical, as usually the service provider only provides the api for the service. I was wondering how the model performs under the black box setting.

The paper mainly presents the experimental results, but lacks of some theoretical insights why intuitively it would work.

**Questions:**

See the weakness. Besides, I was wondering how the initial stage affects the attack results, i.e., say we have total budget on calling gradient of $L$, then to best use this method, how should we trade off the $t_{\textrm{init}}$ and $t_{\textrm{radial}}$.

In figure 2, it shows that the method does not work well on fixup resnet. I was wondering if the authors could conduct more experiments on different architectures to have more evidence.

Some typos: for example

Line: above Line 145, should be $\frac{<z, z'>}{||z|| ||z'||},$ and Line 145 starts with `where`.

---

> ### Author Rebuttal · Authors · 2023-08-09
>
> Thanks for your in-depth review. We address your concerns below :
>
> Regarding weakness 1 : This is right, sometimes the service provider does not provide details. However, we often do know enough about the model’s initial layers as they are often based on some available public model. For example, Resnet, used here, has been used to initialize many different CV models. If the CV model is only provided via a service but we know it is Resnet-based, we can attack it with this method. Note that the adversarial examples exhibit good transfer properties (appendix section J) and simply knowing it is Resnet leaks enough information for us to attack it.
>
> Regarding weakness 2 : In the field of theoretical ML, the presence of BN inducing norm concentration is currently a conjecture with some strong partial evidence behind it - as we show in our key citation of Daneshmand et. al. In fact, we believe this work (https://arxiv.org/abs/2106.03970 - Neurips 2021) to be sufficiently motivating from a theoretical POV in our attack as we argue in section 2.2. Due to space constraints in the main paper, we could not explicate ourselves fully - we promise to expand and build on this connection. We would urge the reviewer to look at the orthogonality gap concept in the linked paper which motivates our attack, especially section 4.1 and appendix F which conjectures that the Gaussianization hypothesis extends to networks which are not linear, i.e. the class we consider. **Despite being aware of these theoretical motivations, we did not wish to simply repeat a different paper's contents - and just cited Daneshmand**. For lack of space, these connections were cut from the main text. If you feel it is more appropriate to re-tell this story in the main paper and also cite the paper, we will do this and write out the motivation following the linked paper to explain why the attack actually has a strong partial theoretical motivation already existing in the literature.
>
> To be clear, fully (not partially) proving the Daneshmand et. al conjecture will be a separate work in its own right, and in theoretical ML. We are focusing on the empirical side, and showing that the partial conjecture is enough to create a good attack. But, we want to make it clear that the theoretical motivation does exist, only in a different paper that we have cited. We will summarize its main content in the paper going forward to make this connection very clear.
>
> Regarding questions :
>
> 1) We have performed some $t_{init}$ and $t_{radial}$ ablations, and attach them appropriately.
>
> 2) We have already performed experiments on several other architectures beyond Resnet as well, like ViT, Efficientnet, VGG16, and have put the results in the paper and the appendix. If you could name the *specific* architecture you wish to see that is not already present, that would be helpful - we will provide it.
>
> 3) Thank you for catching the typos, we will certainly correct them !
>
> **Results upon varying the initial and angular loss tradeoff** :
>
> All results were performed on the vanilla Resnets. The difference is expressed as a difference from the original case (20 initial iterations). The # iterations varied below is the # of such initial iterations. Positive values indicate stronger attack. There is no clear trend except that 20 is actually not the optimal value (15 is a bit stronger). The changes in top-1 accuracy appear in the main value and top-5 in parentheses for all these Resnets with $\epsilon = 0.03$ as that $\epsilon$ avoids accuracy going to $0$, making changes clear and visible.
>
> | Resnet | 5          | 10  | 15  | 20 | 25    | 30 |
> |--------|------------|-----|-----|----|-------|----|
> | 18 | -0.5(-0.6) | 0.1(1.3) | -0.2(0.5) | 0.0 | 0.6(-1.4) | 1.7(0.1) |
> | 34 | 1.5(0.4) | -1.5(-0.5) | 0.8(0.3) | 0.0 | -1.3(1.1) | -1.9(1.7) |
> | 50 | 1.9(-1.6) | -1.3(-0.4) | 1.5(1.7) | 0.0 | -0.1(-1.4) | 0.5(-0.5) |
> | 101 | 1.3(-0.4) | 0.1(1.3) | 1.8(-1.0) | 0.0 | -1.1(1.3) | -0.8(1.6) |
> | 152 | 0.6(-0.3) | 1.6(1.1) | 1.2(0.2) | 0.0 | -0.1(-1.3) | -0.3(-0.5) |
>
> Let us know if you have any other concerns. We hope we have addressed your issues regarding this paper appropriately.

---

> > ### Author Response · Authors · 2023-08-17
> > **Follow-up**
> >
> > Hi, please let us know if the above points adequately address all your concerns with this paper. If not, please let us know so that we can respond and clarify for any further issues.

---

> > > ### Comment · Reviewer_7ce2 · 2023-08-18
> > >
> > > Thank you for the reply and the reference about orthogonal gap.
> > >
> > > In the study of $t_{init}$ and $t_{radial}$, could you also add the (0.0) at the column with 20, otherwise it is a bit confusing to have both positive and negative?
> > >
> > > I will keep the score.

---

> > > > ### Author Response · Authors · 2023-08-21
> > > > **Reply to the response**
> > > >
> > > > Certainly, here you go with the updated table :
> > > >
> > > > | Resnet | 5    | 10   | 15   | 20    | 25   | 30   |
> > > > |--------|------|------|------|-------|------|------|
> > > > | 18     | -0.5(-0.6) | 0.1(1.3) | -0.2(0.5) | 0.0(0.0) | 0.6(-1.4) | 1.7(0.1)  |
> > > > | 34     | 1.5(0.4)   | -1.5(-0.5)| 0.8(0.3)  | 0.0(0.0) | -1.3(1.1) | -1.9(1.7) |
> > > > | 50     | 1.9(-1.6)  | -1.3(-0.4)| 1.5(1.7)  | 0.0(0.0) | -0.1(-1.4)| 0.5(-0.5) |
> > > > | 101    | 1.3(-0.4)  | 0.1(1.3)  | 1.8(-1.0) | 0.0(0.0) | -1.1(1.3) | -0.8(1.6) |
> > > > | 152    | 0.6(-0.3)  | 1.6(1.1)  | 1.2(0.2)  | 0.0(0.0) | -0.1(-1.3)| -0.3(-0.5)|

---

### Official Review · Reviewer_V7by · 2023-07-06

**Soundness:** 4 excellent
**Presentation:** 3 good
**Contribution:** 3 good
**Rating:** 6
**Confidence:** 2

**Summary:**

The authors present and evaluate an algorithm to construct adversarial examples without labels by minimizing the cosine similarity between intermediate layers. The authors show that the attack only works with BatchNorm.

**Strengths:**

- The paper presents strong evidence that the attack is successful on multiple datasets and with multiple architectures

- Despite not achieving state-of-the-art attack success rate when compared to attacks with labels, the paper demonstrates that attacks on BatchNorm architectures are possible without knowledge of the labels, which is required by previous methods

- Paper is generally well written and thorough

**Weaknesses:**

- There are some clarity issues regarding the explanation for the attack, see Questions

- The authors make the assumption that norm is concentrated, and make an argument that it should be concentrated based on statistical assumptions about the data, but it seems that this should be verified empirically by plotting the distribution of the norms, rather than relying on an imprecise argument.

- The assumptions aren't verified to be necessary. It would be helpful to include an ablation where the norm concentration is broken.

**Questions:**

- I don't fully understand why the attack relies on concentration of norm. It's true that without concentration of norm then for a latent representation Z that there is no unique "most dissimilar point", but presumably points opposite Z would still be dissimilar without this assumption.

**Limitations:**

Authors address limitations adequately.

---

> ### Author Rebuttal · Authors · 2023-08-09
>
> Thank you for the review. We would first like to note that you state "The authors show that the attack only works with BatchNorm." Empirically, the attack is successful on other architectures - we carry out experiments on Layernorm architectures (ViT).
>
> Regarding weaknesses :
>
> Weakness 1 : Moved to question, and addressed under questions.
>
> Weakness 2 : Our assumption on the concentration of norm is actually tested in the appendix (page 27) in Tables 49 and 50 empirically. While plots can provide visual intuition, statistical measures are more definitive and we show that FixUp resnets do not concentrate to the same degree normal ones do.
>
> Weakness 3 : Please note that Appendix L contains ablations of non-concentrated layers unrelated to batch norm linearly inside the batch normalized architectures already in table 44, and might fit your needs. Another relevant ablation appears in page 12 of the appendix (table 28)
>
> In general, regarding necessity of concentration - it is very difficult to break the norm concentration in a way that does not also change other things about the setup, making it difficult to test this assumption beyond what we have done in appendix L and FixUp. We have provided evidence that FixUp, for instance, breaks the attack and breaks concentration (see also, the above point). While this **could** result from other attributes of FixUp, this criticism can be raised about any way to not make the norm concentrate as well. If we remove batch norm and have no normalization at all, we will break concentration, but also the other things about the model that batch norm provides. As such, we believe appendix L should meet your requirements.
>
> **If you are not satisfied with the ablations provided, you may provide details about what you would like to see in this ablation, we can try to test it and report here in time.**
>
> We would also note that in general, if an attack proves to be successful outside the conditions we think are necessary, i.e. a concentrated norm, it makes the attack *stronger*, not weaker, from a purely empirical point of view as it allows attacking non-normalized models at least some of the time - even though it does mean the causation might be more complicated.
>
> Re - Question : To be clear, we are **not** saying that the attack necessarily fails in the absence of norm concentration. We are saying that a simple, easily available piece of information like "This network uses batch norm" allows us to conclude concentration of norm is likely, and thus guess our attack might work. Whereas, in the absence of such a guarantee, we are simply less certain about its success. We are not saying it will fail in such a case. In other words, if our attack fails, the norm likely is not concentrated, but if the norm is not concentrated, it does not mean the attack is sure to fail. For example, our attack transfers to VGG16 (Appendix, page 16), which is batch-norm-free.
>
> We hope this addresses your concerns with the paper, while clarifying and resolving any outstanding issues.

---

> > ### Comment · Reviewer_V7by · 2023-08-11
> >
> > I had missed these in the appendix, thanks for pointing this out. Having read the rebuttals and the other reviews, I would like to keep my current score, which is to recommend acceptance.

---

> > > ### Author Response · Authors · 2023-08-17
> > > **Thank you for the response**
> > >
> > > Thank you for the reply. We are glad you found the appendix useful and relevant.

---

### Official Review · Reviewer_fVuA · 2023-07-07

**Soundness:** 3 good
**Presentation:** 1 poor
**Contribution:** 3 good
**Rating:** 5
**Confidence:** 3

**Summary:**

This paper proposes an adversarial attack (angular attack) method that does not require any label information and works by only accessing the network's first part (up to a specific layer). The attack is based on the assumption that the BN layer converges and forms a hyperspherical latent space, where an angular loss is applied to guide the direction of the adversarial updates.

**Strengths:**

1. The proposed angular attack requires no labels, and only partial access to the network.
2. The hyperspherical assumption about the geometry of BN makes sense. Both positive and negative results (e.g., angular attack on Fixup ResNet) support this assumption.


**Weaknesses:**

1. Although Figure 2 is straightforward and informative, the other table results, including those in the appendix, are unclear. For example, it is unclear whether the numbers in Table 9 are absolute mean correlation or the fall in absolute correlation. Table 4 and Table 47 in the appendix are identical tables that only report the “max” over seven baseline methods. I think it makes more sense to include the full results of each baseline.
2. There are some missing ablation studies that I think should be included in the experiment section. a. How does the number of iterations affect the method (currently, the experiments only use a default iteration of 40). b. The current angular loss is computed by averaging the loss over the last two layers; I'd like to see more results of computing losses over more layers or just the last layer.


**Questions:**

1. Table 10 (b) shows that the targeted attack performs worse than the original angular attack. Does that mean the proposed attack method cannot be used for targeted attacks (when some label information is available)?

**Limitations:**

See weaknesses.

---

> ### Author Rebuttal · Authors · 2023-08-09
>
> Hi, thank you for the review. We address the points raised as follows :
>
> Weakness 1 : Table 9 denotes the absolute correlation. It is meant to demonstrate that this value falls monotonically.
>
> Yes, Table 4 and 47 are the same, we apologize for this confusion. We have broken out the results of each method in the common rebuttal. We will include this in the appendix. Due to lack of space, we utilized the max approach for the main paper's table.
>
> Weakness 2 : Yes, we chose 40 as this is a commonly used default iteration # across various papers e.g. the original work on PGD. We attach some ablations with different iteration #.
>
> The averaging only occurs over the last 2 layers due to our preference of keeping things simple - we could certainly optimize over the parameter. Our point was to show that even without this optimization, the attack performs well. We attach ablations over this.
>
> Question :
>
> No, it does not mean that we cannot use the label information. It only means that when we randomly pick an instance of a different label, the angular attack often does better. But, the *combination* of label and angular attack can do better than either attack, so having an instance of a different label can still be very helpful. We provide some cases where our loss is provided in ensemble with the targeted loss and these perform better than either.
>
> All ablations below performed on vanilla Resnets $\epsilon=0.03$ as this is where the accuracy fluctuates the most and at a higher epsilon many accuracies are directly going to zero.
>
> **Table of attacks**
>
> In order in which they are cited (appendix O, Line 219-220). Each value in the column represents added delta from lowest (i.e. gap by which a method is worse). All tests on $\epsilon = 0.03$. Top-1 in main, Top-5 in brackets. "-" denotes optimal.
>
> | ResNet |          |          |          |          |          |          |          |
> |--------|----------|----------|----------|----------|----------|----------|----------|
> | 18     | - | 6.8(2.23) | 6.97(6.54)| 3.96(2.33)| 3.6(2.49) | 1.08(1.1) | 3.78(2.55)|
> | 34     | 3.9(6.31)| 3.07(3.36)| 3.03(1.73)| 2.26(3.2) | 1.95(1.96)| 2.77(4.63)| -  |
> | 50     | 0.32(0.04)| 5.55(7.57)| -  | 1.97(0.33)| 3.58(4.16)| 0.63(7.49)| 2.37(4.65)|
> | 101    | 3.18(6.04)| 3.27(3.6) | 5.55(6.22)| 7.55(8.65)| 1.05(1.68)| -  | 3.14(5.01)|
> | 152    | 3.34(6.15)| 2.45(3.64)| 3.24(5.28)| 1.18(1.95)| 6.91(9.91)| 1.06(2.62)| -  |
>
> **Table of iterations ablated**
>
> Negative values indicate worse performance relative to baseline (40 iterations). We can see there is some gain in moving to 50 iterations. However the standard evaluation practice is with 40, so we did not optimize over this step. As above, we use vanilla Resnets, Imagenet, $\epsilon = 0.03$, top-1(top-5 in bracket) and so on.
>
> | ResNet |        30      |    40          |    50          |
> |--------|--------------|--------------|--------------|
> | 18     | -0.52(-0.43) | 0.0(0.0)     | 0.38(1.97)   |
> | 34     | -1.36(-0.7)  | 0.0(0.0)     | 0.3(0.87)    |
> | 50     | -1.16(-0.34) | 0.0(0.0)     | 0.28(0.02)   |
> | 101    | -0.21(-0.02) | 0.0(0.0)     | 0.63(0.74)   |
> | 152    | -1.56(-0.19) | 0.0(0.0)     | 0.11(1.86)   |
>
> **Number of averaging over layers**
>
> As above, negative values are worse relative to baseline (2 layers to be averaged over). Vanilla Resnets on Imagenet, $\epsilon = 0.03$ , $40$ iterations, top-1 with top-5 in brackets.
>
> | ResNet |      1         |    2      |     3           |        4      |
> |--------|---------------|----------|--------------|--------------|
> | 18     | -1.37(-1.21)  | 0.0(0.0) | 1.09(1.79)   | 3.49(2.93)   |
> | 34     | -2.64(-3.08)  | 0.0(0.0) | 2.95(2.67)   | 3.47(0.93)   |
> | 50     | -2.79(-2.95)  | 0.0(0)   | 2.8(2.15)    | 3.8(3.18)    |
> | 101    | -3.27(-3.55)  | 0.0(0.0) | 0.47(0.31)   | 0.39(0.5)    |
> | 152    | -2.74(-2.98)  | 0.0(0.0) | 0.66(0.45)   | 0.83(0.33)   |
>
> We can clearly see the 2-layer averaging is actually quite helpful. Bigger gains are possible with $3,4$ - but at the cost of more evaluations.
>
> **Ensembling**
>
> These gains are as above - on vanilla Resnets, $\epsilon=0.03$, Imagenet, 40 iters, top-1 with top-5 in brackets. Positive indicates gains. We simply combine the targeted and the angular loss to produce these results.
>
> | ResNet |               |
> |--------|---------------|
> | 18     | 3.38(1.62)  |
> | 34     | 0.88(2.05)  |
> | 50     | 0.64(3.81)  |
> | 101    | 0.89(1.79)  |
> | 152    | 0.99(2.1)  |
>
> Please note that a similar ablation with PGD also appears in appendix (Appendix N).
>
> We hope that our rebuttal answers all your queries to your satisfaction and resolves your issues with the paper.

---

> > ### Comment · Reviewer_fVuA · 2023-08-13
> >
> > Thank the authors for their response. The additional results answered my questions, and I am happy to raise my score.
> >
> > However, I’d like to add that ablation studies are essential to show how sensitive the proposed method is to different hyper-parameters and help readers to understand the method better, so I can't entirely agree with the author’s argument on why they didn’t conduct them in the first place.
> > In addition, I agree with Reviewer ykjk that the presentation of this paper should be improved. Currently, the numbers in the tables (including those in the rebuttal) are not straightforward to understand (I personally don't think it's a good idea to only report the "delta" value and not include the baseline number. For example, if the baseline number is small, then a small delta means a large difference.). It’s also hard to compare different methods because they are not in the same table.

---

> > > ### Author Response · Authors · 2023-08-17
> > > **Reply to the reviewer**
> > >
> > > We are glad to hear the rebuttal addresses your concerns and also glad that you have raised your score, thank you !
> > >
> > > Yes, we should have clarified this point. The ablation study should show we have not cherry-picked, as was our initial intent behind fixing the hyperparameters.
> > >
> > > Due to the large number of models being compared, it is hard to convey the figures all in one table. But we will certainly make efforts in this direction. The delta values were used over raw values as we felt the delta values were clearer...but if the reverse is true, we will go back to raw values.

---

### Author Rebuttal · Authors · 2023-08-10

Dear Reviewers,

We thank you for your reviews, and have directly responded to all of you without any common rebuttal. Please let us know if your concerns are appropriately addressed fully.

---

### Decision · Program_Chairs · 2023-09-21

**Decision:**

Accept (poster)

**Comment:**

The paper presents a method for creating adversarial examples using batch normalization without relying on labels. After carefully considering the reviewers' comments and the authors' rebuttals, I recommend accepting this paper for the following reasons:

Theoretical Soundness and Experimental Promises: As highlighted by several reviewers, the proposed method is technically sound and well-motivated (Reviewer ykjk: "The proposed method is well motivated and technically sound"). The experimental results are promising, with successful attack rates on different datasets and architectures (Reviewer V7by: "The paper presents strong evidence that the attack is successful on multiple datasets").

Response to Concerns: The authors effectively addressed most of the reviewers' concerns in the rebuttal. They provided additional results, clarified misunderstandings, and committed to improving the presentation. For example, Reviewer fVuA's concerns about ablation studies and table clarity were addressed, leading to the statement: "The additional results answered my questions, and I am happy to raise my score."

Significance of Contribution: Although there were concerns about the practical applicability and significance of the method, the authors clarified its relevance in scenarios involving pre-trained and fine-tuned models (Authors: "The proposed method is most suitable for the cases where a feature extractor (known to us) has been used to train a model (unknown to us)").

Limitations and Presentation: Concerns about clarity, presentation, and comparison with existing methods like PGD were acknowledged by the authors. They have promised to address these in the final version, and their detailed explanations seem to support this commitment (Authors: "We will put all this in the paper in terms of content").

While one reviewer maintained a Borderline reject rating (Reviewer ykjk), the overall consensus leans toward acceptance, and the authors have provided satisfactory responses to many of the raised issues.

In conclusion, the paper's theoretical grounding, experimental support, and the authors' diligent response to the reviewers' concerns lead to a recommendation for acceptance.